# Measuring the intrinsic dimension of Earth representations

**Arjun Rao**[*]
Department of Computer Science
University of Colorado Boulder

**Marc Rußwurm**
University of Bonn &
Wageningen University

**Konstantin Klemmer**
LGND AI &
University College London

**Esther Rolf**
Department of Computer Science
University of Colorado Boulder

## Abstract

Within the context of representation learning for Earth observation, geographic Implicit Neural Representations (INRs) embed low-dimensional location inputs (longitude, latitude) into high-dimensional embeddings, through models trained on geo-referenced satellite, image or text data. Despite the common aim of geographic INRs to distill Earth's data into compact, learning-friendly representations, we lack an understanding of how much information is contained in these Earth representations, and where that information is concentrated. The intrinsic dimension of a dataset measures the number of degrees of freedom required to capture its local variability, regardless of the ambient high-dimensional space in which it is embedded. This work provides the first study of the intrinsic dimensionality of geographic INRs. Analyzing INRs with ambient dimension between 256 and 512, we find that their intrinsic dimensions fall roughly between 2 and 10 and are sensitive to changing spatial resolution and input modalities during INR pre-training. Furthermore, we show that the intrinsic dimension of a geographic INR correlates with downstream task performance and can capture spatial artifacts, facilitating model evaluation and diagnostics. More broadly, our work offers an architecture-agnostic, label-free metric of information content that can enable unsupervised evaluation, model selection, and pre-training design across INRs.

## 1 Introduction

Across vision, audio, and other modalities, seemingly high-dimensional observations often vary along far fewer degrees of freedom. This is especially true of geographic data, which is often characterized by strong spatio-temporal dependencies. For example, classical work in meteorology uses dimensionality reduction techniques since large-scale oscillations in climate trends can be explained by a handful of indices (van den Dool, 2006). This phenomenon is leveraged by a class of representation learning techniques aimed at embedding signals in Earth's data into succinct, general purpose vector representations (Rolf et al., 2025). This is done either through direct embedding of geo-referenced data with image or text encoders, or through a new class of geographic implicit neural representations (INRs) that encode geospatial signals in the weights of a location encoder network which takes geographic position (latitude and longitude) as input.

Currently, the quality of Earth representation models is evaluated largely in terms of supervised model performance for specific downstream tasks. Pre-trained geographic INRs have driven state-of-the-art performance in tasks like land cover segmentation, object detection, and image geo-localization (Cepeda et al., 2023; Klemmer et al., 2025; Mai et al., 2023a; Liu et al., 2025). In addition, geographic INRs are increasingly used to improve geospatial data interpolation (Mac Aodha et al., 2019a; Chen et al., 2025; Lange et al., 2025), for instance, in global species distribution mod-

---

[*]Corresponding authors: `raoarjun@colorado.edu` or `marc.russwurm@uni-bonn.de`

Figure 1: **Estimating the intrinsic dimension (ID) of geographic implicit neural representations (INRs).** We compute the ID of geographic INRs in two ways, to measure model representativeness and task-alignment. **Representativeness** (left): We generate location embeddings with frozen pre-trained location encoders for coordinates across Earth's landmass. We calculate the global and local ID values on the resulting embeddings. **Task-alignment** (right): We train a downstream task-specific model using location embeddings as input. We use a TwoNN ID estimator to measure the ID of the activations of the task-specific model's last hidden layer.

eling (Cole et al., 2023b; Dhakal et al., 2025). While task-specific metrics evaluate the "learning-friendliness" property of location encoder models (as outlined by Mai et al. (2022)), focusing only on task-specific metrics prevents us from measuring progress on a fundamental aim of location embeddings: to generate rich, general purpose representations of Earth's data.

In this work, we study the intrinsic dimension (ID) of geographic INRs as a task-agnostic (unsupervised) metric to quantify information-richness across space and where that capacity is concentrated. Defined in Section 2.2, the intrinsic dimension measures how many independent directions a learned representation actually uses. We compute point-wise, *local* estimates of ID to capture regional effects as well as *global* ID metrics for comparison between different location encoder models. Our results show that the intrinsic dimension reveals two important, and previously unexplored properties of geographic INRs: (i) **representativeness** – the amount of independent, non-redundant variation of the INRs and (ii) **task-alignment** – indicating how well downstream predictors can compress the INRs onto a low-dimensional, target-aligned manifold. Our overall methodology is summarized in Figure 1 and our key findings can be summarized as follows:

- Global ID estimates of current geographic INRs are an order of magnitude lower than their ambient size, yet are competitive with ID estimates of Earth embeddings generated with large-scale image encoders.

- ID estimates of geographic INRs increase with additional input modalities and increased spatial resolution of the location encoder, highlighting that ID captures increases in spectral and spatial representativeness of these embeddings.

- Local ID estimates reveal spatial artifacts of pre-trained geographic INRs, which can arise from biases in the pre-training dataset coverage or properties of their model architectures.

- Global ID correlates with downstream task performance across encoders and tasks. Interestingly, correlation is positive when ID is calculated in the embedding space of frozen, pre-trained models, but is negative when ID is calculated in the activation space of supervised downstream models. This sheds light on the connections between the intrinsic dimension, representativeness, and task-alignment of geographic INRs.

While much of our analysis extends naturally to general classes of INRs, geographic INRs are a particularly interesting case to study because the underlying domain geometry is explicit. Inputs lie on the sphere ($S^2$), which makes it possible to separate the known 2D manifold from learned information content above it. Moreover, the explicit goal of many geographic Earth embedding models is to compress and coalesce as much signal from Earth's data as possible. The strategies we present for estimating different types of ID offer a new unsupervised evaluation strategy for representation learning of Earth data. Our code is available at https://github.com/arjunarao619/GeoINRID.

## 2 RELATED WORK

### 2.1 GEOGRAPHIC INRS AND LOCATION ENCODERS

Implicit neural representations (INRs) encode signals as functions that map coordinates to values using neural networks (Chen & Zhang, 2019; Mildenhall et al., 2020). Analogously, geographic INRs encode geographic signals on the sphere using location encoder networks. These models take geographic coordinates (longitude, latitude) as input and return a corresponding value or vector embedding (Mai et al., 2022). Location encoders can be trained directly using a supervised loss to, for instance, interpolate between species observations (Cole et al., 2023b) or sea ice thickness measurements (Chen et al., 2025). Alternatively, they can be (pre)trained on unlabeled data, e.g., using a contrastive objective, to obtain an *embedding vector*.

Location encoder architectures consist of a positional encoding (PE), projecting longitude/latitude inputs into a higher-dimensional feature space, and a neural network projecting these features into the desired output space. Popular positional encodings include multi-scale sinusoidal functions (e.g. Sphere2Vec (Mai et al., 2023b) and Space2Vec (Mai et al., 2020)), multi-scale Random Fourier Features (RFFs) as used in GeoCLIP (Cepeda et al., 2023), and spherical harmonic functions that provide an orthogonal, sphere-native basis (Rußwurm et al., 2024). The "resolution" of the location embeddings is controlled by these positional encoding hyperparameters. The most common pre-training objective for location encoders is contrastive image-location matching. This way, location-specific image features are encoded in the location encoder network and—at inference time—can be accessed by providing solely coordinate inputs. Examples include SatCLIP (Klemmer et al., 2025) and GeoCLIP (Cepeda et al., 2023), which are available as pre-trained models and can be seamlessly integrated in other frameworks, for instance, in location-aware image synthesis (Sastry et al., 2024) or super-resolution (Panangian & Bittner, 2025).

An alternative way to obtain location embedding vectors is to download raw data (e.g., a satellite image) at one location and use a pre-trained vision model as an image encoder. Single-modality pre-trained ResNets and ViTs are available in repositories like TorchGeo (Stewart et al., 2025) and SSL4EO (Wang et al., 2023). Recent work on multi-modal remote sensing foundation models like DOFA (Xiong et al., 2024), CROMA (Fuller et al., 2023), or MMEarth (Nedungadi et al., 2024) are other large-scale geospatial image encoders. In this work, we primarily study the intrinsic dimension of pre-trained location encoders as continuous geographic INRs, but also compare them quantitatively to embeddings from image encoders.

### 2.2 INTRINSIC DIMENSION (ID)

The intrinsic dimension of a dataset can be thought of as a nonlinear analogue of Principal Component Analysis (PCA). While PCA finds a single global linear rank, ID captures the minimal number of degrees of freedom needed to describe data locally. ID is uniquely suited for measuring the true dimensionality of data representations since they occupy a curved manifold (Ansuini et al., 2019). Intrinsic dimension has been used in deep learning in several ways: to define a normalized notion of task difficulty (Li et al., 2018), explain generalization ability and sample efficiency (Pope et al., 2021; Ansuini et al., 2019; Gong et al., 2019), characterize adversarial examples (Ma et al., 2018), detect AI-generated content (Lorenz et al., 2023; Tulchinskii et al., 2023), or to regularize local or joint input–feature dimensions to improve self-supervised representations (Huang et al., 2024; Zhu et al., 2018). Learning theory treats intrinsic dimensionality as a key factor influencing learnability (Narayanan & Niyogi, 2009; Narayanan & Mitter, 2010).

#### 2.2.1 ESTIMATORS OF ID

ID estimators quantify how the local neighborhood around an embedding $z \in \mathbb{R}^D$ grows within a small ball in feature space. The ID can be calculated with *distance-based* or *angle-based* estimators. Distance-based methods assume that, in a sufficiently small neighborhood of $z$, samples are drawn from an approximately uniform Poisson process on a $d$-dimensional manifold. Let $R_1(z) \leq \cdots \leq R_k(z)$ be the Euclidean distances to the $k$ nearest neighbors of $z$. The Levina–Bickel maximum

likelihood estimator (MLE) (Levina & Bickel, 2005) defines a local ID estimate

$$\hat{d}_{\text{MLE}}(z) = \left[ \frac{1}{k-1} \sum_{j=1}^{k-1} \ln \frac{R_k(z)}{R_j(z)} \right]^{-1},$$

and aggregates these to a global ID via a harmonic mean over embeddings $z_i$. The TwoNN (Facco et al., 2017), MOM (Amsaleg et al., 2018), and TLE estimators are natural variants of this same Poisson–ball model. They replace the full set of log-ratios $\{\ln(R_k/R_j)\}$ by alternative summaries of the $k$-NN configuration (for example, the first two radii or the mean radius $\bar{R}(z)$ relative to $R_k(z)$), but still infer $d$ from how quickly the number of neighbors grow with distance.

Angle-based estimators instead operate on directions after re-centering and whitening. FisherS (Albergante et al., 2019) standardizes the embeddings $\{z_i\}_{i=1}^n$ by mean-centering, retaining well-conditioned principal components, whitening them, and projecting each point onto the unit sphere to obtain $x_i = z_i / \|z_i\|$. At a cosine similarity margin $\alpha \in (0,1)$, it measures an empirical inseparability frequency $\bar{p}(\alpha)$, defined as the fraction of pairs $(i,j)$ with $\langle x_i, x_j \rangle > \alpha$. For points sampled uniformly on a unit $n$-sphere, this frequency is well-approximated by

$$\bar{p}_{\text{sphere}}(\alpha; n) \approx \frac{(1-\alpha^2)^{(n-1)/2}}{\alpha \sqrt{2\pi n}},$$

and FisherS inverts this reference curve to recover an effective dimension

$$\hat{n}(\alpha) = \frac{W\left( -\dfrac{\ln(1-\alpha^2)}{2\pi \, \bar{p}(\alpha)^2 \, \alpha^2 (1-\alpha^2)} \right)}{-\ln(1-\alpha^2)},$$

where $W(\cdot)$ is the Lambert $W$ function. Since angle-based estimators infer ID from the angular spread of neighbor directions by recentering the point's neighborhood and using only the unit directions to its neighbors, they are robust to local spatial variabilities.

To our knowledge, no prior work measures the intrinsic dimensionality of implicit neural representations—especially *geographic* INRs; our study is the first to provide these measurements and link ID to generalization and representativeness in geographical settings.

## 3 ESTIMATING THE ID OF GEOGRAPHIC INRS

We model a pre-trained location encoder as a map $f : S^2 \to \mathbb{R}^D$ that returns an embedding $z = f(x)$ for a geographic location $x = (\lambda, \phi)$. The intrinsic dimension at $x$, which we denote as $d(x)$, summarizes how many independent directions the embedding $z$ varies when we perturb $x$ slightly on Earth's surface. Equivalently, $d(x)$ is the smallest number of coordinates needed to describe the support of the embedding distribution in a small neighborhood of $z$. The ambient dimension $D$ of the embedding is fixed by the last layer of the location encoder architecture, whereas $d(x) \leq D$ reflects how much distinct geographic signal the encoder actually expresses at that location.

We estimate the geographic INR ID across two scales: The **local ID** $d(x)$ reveals where the representation is complex or compressive across Earth's surface, while the **global ID** aggregates $d(x)$ over a specified set of locations to provide a single scalar value. We report both: local ID maps can diagnose spatial heterogeneity and the global ID allows us to compare between location encoders. High local ID values signal regions where embeddings vary along many independent directions whereas low values reveal compressive regimes where the representation is effectively one- or two-dimensional.

The choice of angle- versus distance-based ID estimators is also important in our analysis: Angle-based estimators' robustness to spatial heterogeneity make them a natural choice to estimate a *global* intrinsic dimension over the Earth's surface. For global analyses, distance-based estimators can be disproportionately biased by local patterns as they read changes in spacing as changes in dimension. Within the context of heterogeneous representations of the Earth, these local patterns can be induced by changing climate zones or terrain edge effects, for instance, in coastal areas. However, this sensitivity can be beneficial in *locally* analyzing where these intrinsic dimensions change spatially. Thus, our measurements of local ID use distance-based estimators.

### 3.1 MEASURING REPRESENTATIVENESS AND TASK-ALIGNMENT WITH ID

As illustrated in Figure 1, we estimate intrinsic dimension (ID) at two different stages:

1. **Measuring representativeness in embedding space:** With a frozen location encoder $f$ and *globally* sampled $N$ geographic coordinates $(\lambda, \phi)$, we form embeddings $Z_{\text{geo}} = f(\text{PE}(\lambda, \phi)) \in \mathbb{R}^{N \times D}$ and estimate the global and local ID of these embeddings. When correlating global ID to downstream task performance, we use the angle-based FisherS estimator.

2. **Measuring task-alignment in activation space (dataset–conditioned):** We then train a shallow classifier on task-specific locations and compute ID with the distance-based TwoNN estimator on the penultimate ReLU activations evaluated only at the dataset's coordinates for each split. Concretely, for the train/validation/test splits we pass their spatial coordinates through $f$, feed the resulting embeddings to the classifier, and estimate TwoNN ID jointly for the full dataset. This follows established practice of measuring the ID of neural network representations first introduced in Ansuini et al. (2019).

Unlike other interpretations of task-alignment that use statistical dependence metrics such as mutual information and estimate task-relevant dependence between embeddings and labels via variational bounds (Cheng et al., 2020; Hjelm et al., 2019; Liu et al., 2024), our notion of task-alignment is explicitly geometric. We say that a representation is task-aligned when a shallow head can compress it onto a low-dimensional manifold, which we probe directly through the intrinsic dimension of its activations.

### 3.2 DATASETS AND EXPERIMENTS

In our experiments, we correlate the estimated ID of different geographic INRs (detailed in section 2.1) with downstream task performance on several geospatial regression and classification tasks that use either (i) only location $(\lambda, \phi)$ context or (ii) location and additional context (e.g. an image). Location-based regression tasks include air temperature (Hooker et al., 2018), and elevation, population density, nightlights, and tree-cover prediction from Rolf et al. (2021). Location-based classification tasks include biome and countries classification from Klemmer et al. (2025). We also measure the ID-performance relationship on several image-location regression tasks aimed at measuring socioeconomic outcomes from the SustainBench (Yeh et al., 2021) benchmark. We use the task setup, including labels and precomputed InceptionV3 image features provided through the TorchSpatial benchmark for location encoders (Wu et al., 2024).

Our downstream classification/regression heads trained on these tasks are either a 2 or 3-layer MLP. For the SustainBench image-location regression tasks, we use one 2-layer MLP branch that receives image features as input, and a 5-layer MLP which takes the geographic coordinates as input. All fine-tuning experiments are trained for between 20-50 epochs with an early-stopping condition on a held-out validation loss. We use a grid search with the Optuna framework (Akiba et al., 2019) to determine the optimal learning rate, best hidden dimension size of the MLP, and best values for weight decay. Mean performance metrics across 10 random seeds are reported.

When measuring the effect additional input modalities have on geographic INR ID and representativeness (Section 4.3), we use MMEarth, a multi-modal Earth observation corpus with Sentinel-2 optical, Sentinel-1 SAR, ASTER GDEM terrain, ETH-GCHM canopy height, Dynamic World land cover, and ESA WorldCover data. Within SatCLIP we compare three variants: (i) an S2-only baseline with a MoCo-pre-trained ResNet-50 image encoder trained on MMEarth's Sentinel-2, (ii) a SatCLIP location encoder pre-trained on Sentinel-1 and Sentinel-2 imagery, and (iii) SatCLIP pre-trained on all MMEarth pixel-level modalities. For each model we compute global FisherS ID on uniformly sampled land-only coordinates, then train a small 3-layer MLP on frozen embeddings to predict air temperature, elevation, and population-density values.

## 4 RESULTS

### 4.1 GLOBAL AND LOCAL ID OF GEOGRAPHIC INRS

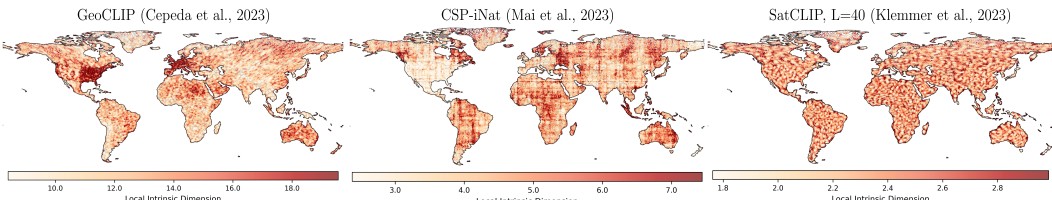

Figure 2: **Local intrinsic dimension of geographic INRs reveal spatial artifacts.** We use the MLE estimator on embeddings generated over Earth's landmass. $N = 100,000$ points sampled with $k = 100$ neighbors used in the MLE ID calculation. We plot the local ID of more INRs in Appendix Figure 11.

**Global geographic INR ID is significantly lower than its ambient dimension but often greater than 2.** Table 1 presents global ID estimates for geographic INRs from pre-trained location encoders and commonly used geospatial image encoders. We observe that the global IDs are substantially lower than the ambient embedding dimensions ($D$) for all geographic INRs. SatCLIP and CSP embeddings are 256-dimensional while GeoCLIP uses 512 dimensions yet all ID estimates for these models fall below 14. In Appendix Table 2, we empirically verify the reliability of ID over other dimensionality reduction techniques such as PCA and ICA. We observe that the number of retained components in both PCA and ICA sharply rise with increasing $D$, while ID remains relatively stable. This is verified in Appendix Table 2b where we observe that increasing the $D$ in SatCLIP does not accompany with it an improvement in task performance.

The ID varies substantially by location encoder architecture. GeoCLIP shows IDs of 11-13 across distance-based estimators, CSP variants range from 3-6, while SatCLIP remains lowest at 2-2.5. Estimator choice also matters: Angle-based FisherS produces notably different patterns—yielding 8.08 for SatCLIP-L40 (versus 2-2.4 for distance-based methods MLE, MOM, TLE) but drops below 2 for CSP models.

Table 1: **Global intrinsic dimension of Earth embeddings.** Distance-based estimators use $k = 20$ nearest neighbors. Appendix Table 3 shows results for more estimators and sampling schemes.

| Model | $D$ | FisherS | MLE | MOM | TLE |
|---|---|---|---|---|---|
| *Location encoders, Land sampling (100k points)* | | | | | |
| SatCLIP–L10 | 256 | 5.00 | 1.96 | 2.02 | 2.16 |
| SatCLIP–L40 | 256 | **8.08** | 2.03 | 2.39 | 2.32 |
| GeoCLIP | 512 | 7.68 | **11.21** | **13.02** | **11.53** |
| CSP–fMoW | 256 | 1.70 | 5.18 | 5.23 | 6.25 |
| CSP–iNat | 256 | 0.92 | 3.37 | 4.64 | 4.14 |
| SINR | 256 | 3.19 | 2.19 | 3.36 | 2.74 |
| TaxaBind-Loc | 512 | 3.33 | 9.44 | 11.56 | 10.30 |
| *Image encoders on S2-100K (Klemmer et al., 2025)* | | | | | |
| RCF | 512 | 1.64 | 6.32 | 5.23 | 7.10 |
| CROMA | 768 | **9.79** | 19.57 | 17.00 | 20.30 |
| DOFA | 768 | 3.32 | 15.58 | 13.78 | 16.20 |
| ResNet18 | 512 | 6.32 | 16.14 | 12.27 | 16.80 |
| ResNet50 | 2048 | 6.42 | 16.27 | 13.18 | 17.00 |
| ResNet152 | 2048 | 7.60 | **20.72** | 17.50 | **21.50** |
| ViT-Small | 384 | 3.33 | 18.53 | 15.80 | 19.20 |
| AlphaEarth | 64 | 4.67 | 7.80 | 9.48 | 8.61 |
| TaxaBind-Sat (RGB) | 512 | 3.39 | 10.04 | 14.98 | 12.80 |
| ScaleMAE (RGB) | 1024 | 2.96 | 10.16 | 8.90 | 11.00 |
| ResNet18 (RGB) | 512 | 0.92 | 10.85 | 8.70 | 11.70 |
| ResNet50 (RGB) | 2048 | 0.92 | 9.92 | 8.10 | 10.80 |
| DINOv3-Sat (RGB) | 4096 | 3.87 | 13.03 | **19.52** | 15.95 |

**Global IDs of geographic INRs are similar to that of embeddings derived from image encoders.** The intrinsic dimension of geographic location encoders (purely with a $(\lambda, \phi)$ context) record similar intrinsic dimension estimates compared to large-scale image encoders evaluated on S2-100K Sentinel-2 tiles (Table 1). GeoCLIP's ID of 11-13 approaches that of foundation models like DOFA (ID: 14-16) (Xiong et al., 2024), CROMA (ID: 17-20) (Fuller et al., 2023), and is higher than the ID of AlphaEarth embeddings (4-9) (Brown et al., 2025), the pre-trained SINR model from Cole et al. (2023a), and TaxaBind's multi-modal satellite image encoder (Sastry et al., 2025). This indicates that current pre-trained location encoders contain a similar amount of overall information content as embeddings of multi-spectral satellite imagery obtained through specialized image encoders, as measured through these global ID estimators.

**Local estimates of ID reveal spatial artifacts of pre-trained geographic location encoders.** Figure 2 plots local ID estimates of pre-trained geographic INRs using the distance-based MLE estimator with $k = 100$ nearest neighbors. (Results are not very sensitive to the choice of $k$, which we further evaluate in Appendix B). For GeoCLIP, local ID is highest in the United States and west-

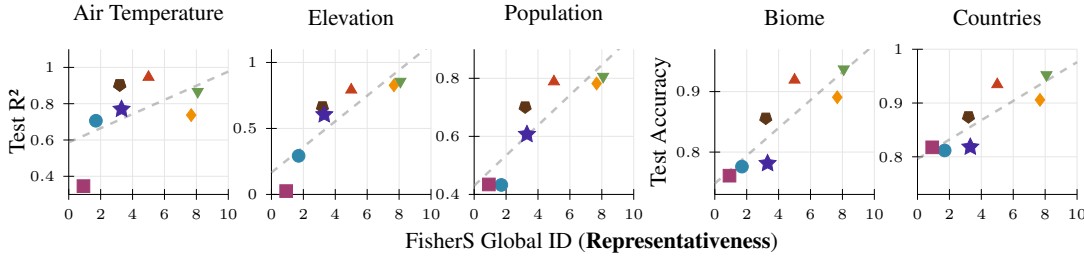

(a) FisherS global ID of geographic INRs vs test $R^2$ and top-1 accuracy using land-based coordinate sampling.

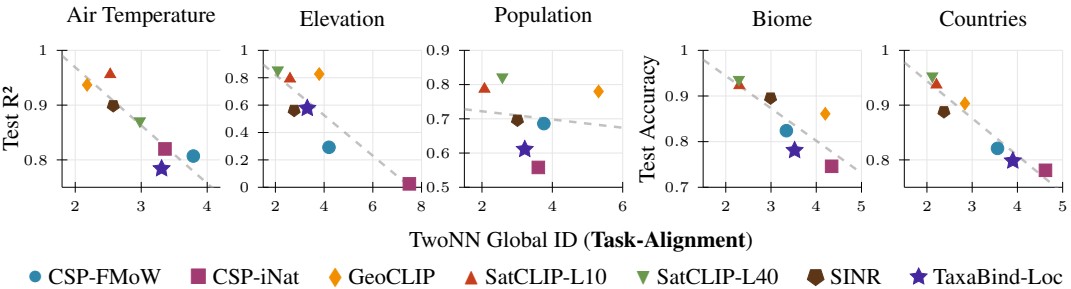

(b) TwoNN global ID of ReLU activations of a 3-hidden-layer MLP's penultimate layer vs test $R^2$ and top-1 accuracy.

Figure 3: **Relationship between global ID of geographic INRs and downstream task performance** measured across five regression and classification tasks. In both rows, the location embeddings are frozen while task-specific predictions heads (3 layer MLPs) are learned. In (a), ID (horizontal axis) is calculated on the frozen pre-trained embeddings as in Table 1. In (b), ID is measured in *activation space* using the TwoNN estimator on a learned classifier's penultimate layer.

ern Europe, reflecting the spatial distribution of the social-media images on which it is pre-trained. The local IDs of CSP show a grid pattern because its positional encoding repeats at regular steps in longitude and latitude, so equally spaced locations look alike and form bands along meridians and parallels. For SatCLIP, local ID maps show no regional coverage bias (consistent with S2-100K's global sampling), but they exhibit thin, periodic oscillations, reflecting the finite-order spherical-harmonic functions used in its location encoder.

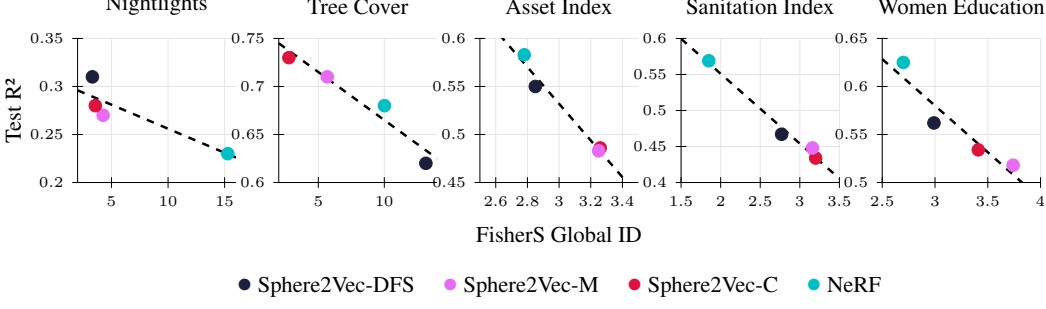

Figure 4: **FisherS global ID of task-specific location embeddings learned via supervised learning vs test $R^2$ of four continuous location encoders on five tasks from TorchSpatial (Wu et al., 2024)** FisherS ID is calculated on the intermediate location embeddings from the location encoder, similar to Figure 3a. The asset index, sanitation index, and women education tasks are image-location regression tasks, as detailed in Section 3.2.

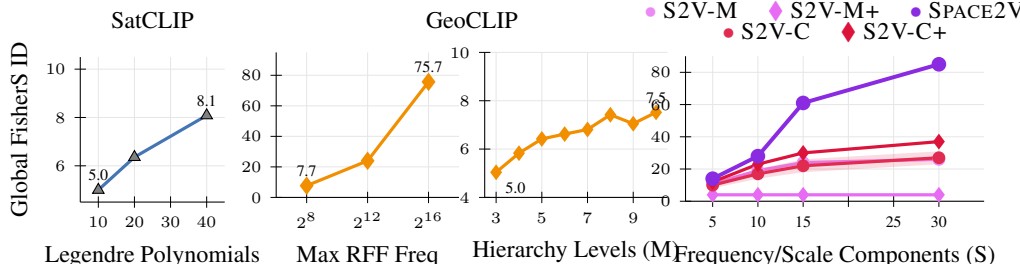

Figure 5: **Effect of location encoder spatial resolution on global ID.** (Left) for SatCLIP, we pre-train the location encoder with $L = 10, 20$, and $40$ Legendre Polynomials. (Middle) For GeoCLIP, we increase both the maximum RFF frequency and the number of hierarchical levels (M) used by the location encoder by fine-tuning the new higher-frequency branches on a YFCC (Thomee et al., 2016) image geo-localization task. (Right) For Sphere2Vec (S2V) and Space2Vec (SPACE2V) encoders, we increase the number of frequency components (S) and train the location encoder with supervised learning on the MOSAIKS nightlights regression task.

## 4.2 GEOGRAPHIC INR ID CORRELATES WITH TASK PERFORMANCE

**Geographic INRs with high global ID record higher task performance.** In Figure 3a, we compute the *global* FisherS ID in embedding space to measure representativeness as pictured in the left panel in Figure 1. We then correlate these ID estimates with the performance of task-specific supervised learning models (small MLPs trained on top of embeddings from frozen geographic INRs). Across datasets and tasks, the scatter plots exhibit a clear positive linear trend: geographic INRs with larger global ID lead to higher downstream performance. A higher global ID implies a richer coverage of geographic variability—i.e., more task-relevant directions are available for a shallow learner to exploit with limited supervision.

**Lower global ID in activation space of supervised models corresponds to higher task performance.** Still keeping the geographic INR frozen, we train a small task head and then measure the global TwoNN ID in activation space of this downstream task head (following the experimental settings in Ansuini et al. (2019)). This measures task-alignment as pictured in the right panel in Figure 1. In Figure 3b, we observe a strong negative correlation between activation-space ID and performance, indicating that supervised adaptation compresses the INR features onto a task-aligned manifold with fewer effective degrees of freedom. This is consistent with past work, that found lower ID indicates more concentrated, linearly separable structure and thus better generalization (Ansuini et al., 2019; Pope et al., 2021; Zhu et al., 2018).

**Within a supervised learning setting, lower ID of task-specific location encoders corresponds to higher task performance.** To further investigate task-alignment, we train continuous location encoders Sphere2Vec and Space2Vec end-to-end with supervised learning on the image-location regression tasks outlined in Section 3.2. We then compute the global FisherS ID on the task-specific learned embeddings, across training sample locations. In Figure 4, interestingly, we again find a consistent negative relationship between ID and $R^2$, suggesting that direct supervision drives the representations toward a lower-ID, task-specific manifold that is easier to separate or regress on. This mirrors Figure 3b and reinforces the picture that the benefits of self-supervised pre-training stem from *high* global ID (expressivity/coverage), while subsequent supervised models benefit from *low* global ID (compression/task-alignment).

## 4.3 EFFECT OF RESOLUTION AND INPUT MODALITIES ON THE ID OF GEOGRAPHIC INRS

Having established that ID measures can capture the relative information content of geographic INRs across space and relate to downstream task-specific performance, we now examine how ID values change when the properties of geographic location encoders change. We focus on two properties along which geographic location encoders are routinely modified that should result in different information content in their embeddings: the spatial resolution of the location encoder architecture, and the input modalities used during pre-training.

**Increased spatial resolution of geographic INRs increases global ID and representativeness.**
The spatial resolution of several location encoder architectures is controlled by specific model hyperparameters: spherical harmonics by the number of Legendre polynomials used $L$, Random Fourier Features (RFF) by their maximum frequency $\sigma_{\max}$ and hierarchy depth $M$, and Space2Vec/Sphere2Vec via the number of multi-scale components $S$. In Figure 5, we plot how ID changes as we vary these hyperparameters. Intuitively, increasing resolution should allow the encoder to resolve finer geospatial phenomena and thus use more independent directions in representation space. For SatCLIP (equipped with a spherical harmonic and sinusoidal representation network positional encoder), the global FisherS ID rises with $L$. Similarly, for GeoCLIP, increasing $\sigma_{\max}$ by appending high–frequency branches to the original pre-trained set and increasing the hierarchy density $M$ both increase global ID. Increasing $\sigma_{\max}$ produces a sharp increase in global ID, whereas increasing $M$ produces more gradual increases. Across Space2Vec variants, as we vary the $S$ parameter, the rise in ID is steepest for the theory/grid formulation of Space2Vec, moderate for the compositional variants, and smallest for the multiplicative variants. This supports the view that adding

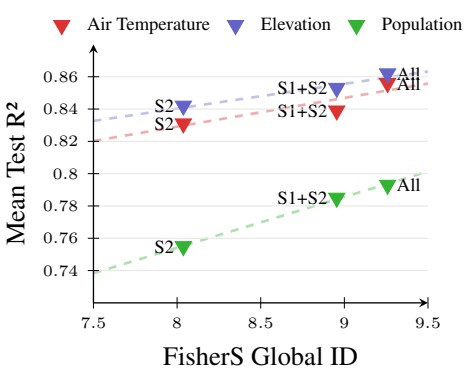

Figure 6: **Using additional input modalities during pre-training increases ID and downstream task performance of geographic INRs.** Colors represent different tasks. Models are pre-trained on subsets of the MMEarth dataset: Sentinel-2 (S2), Sentinel-1 and 2 (S1+S2), and all available rasters (All).

multi-scale components enlarges the set of independent directions the encoder can use, thereby increasing the representational capacity of the location embedding.

**Additional pre-training data modalities increases global ID and representativeness.** In Figure 6, we train SatCLIP models with different subsets of pre-training data modalities from the MMEarth dataset. We find that increasing the number of geographic layers seen during training increases both the global ID and its downstream task performance. SatCLIP location encoders with the most number of input modalities (optical Sentinel-1, multispectral Sentinel-2 and all pixel-level modalities on MMEarth) record both the highest global ID and performance across an air temperature, elevation, and population density regression task. This confirms that ID can capture differences in information content due to increasing the number modalities beyond multi-spectral Sentinel-2 imagery. The clear gains to downstream performance in Figure 6 echoes our results linking increased representativeness to increased downstream task performance in Figure 3.

## 5 DISCUSSION AND CONCLUSION

Intrinsic dimension (ID) offers an architecture- and task-agnostic metric to measure the information content encoded in geographic INRs. Despite being a task-agnostic metric, global ID still informs the "learning-friendliness" (Mai et al., 2022) property of a location encoder. When measured on frozen embeddings, higher global ID aligns with stronger downstream performance after fine-tuning, indicating greater representativeness. When measured on the activations of a supervised prediction head, lower global ID accompanies better generalization, consistent with task-aligned compression. Local ID maps expose spatial artifacts that could influence the performance of geographic INRs on downstream applications. Thus, both global and local ID analysis can support model selection via label-free model evaluation at the pre-training stage.

Beyond this model selection role, our results suggest several concrete applications of ID for designing, auditing, and deploying geospatial machine learning models. First, global ID of frozen location encoders can act as a label-free proxy for downstream performance when comparing architectures, positional encodings, resolution hyperparameters, or input modality sets, reducing the cost of exploring new INR designs. Second, monitoring the divergence between encoder and head ID offers a signal for early stopping that complements standard validation metrics. Third, spatially resolved local ID maps allow practitioners to audit geographic biases, guiding targeted data collection and region-aware fine-tuning.

While these applications already make ID practically useful, we view it as one component of a broader evaluation toolkit for geographic representation learning. Beyond the supervised regression and classification benchmarks we study here, ID-based diagnostics could be extended to geo-prior settings such as location-conditioned satellite image generation and fine-grained classification with geospatial priors (Feng et al., 2025; Mac Aodha et al., 2019b). Complementary measures could quantify data provenance by attributing information content to specific pre-training corpora or regions, and could improve interpretability by localizing that content to subspaces of the embedding. Future work could also develop more principled ways to use patterns in local ID to drive task-specific fine-tuning or data acquisition strategies that adapt to regional variation in information content.

## ACKNOWLEDGEMENTS

A majority of training runs conducted in this work were run on an NVIDIA Grace-Hopper (GH200) GPU node provided by the University of Colorado Boulder's high performance computing system Alpine. We thank Brandon Reyes and the RC computing team at CU Boulder for allowing priority access to this resource. Alpine is jointly funded by the University of Colorado Boulder, the University of Colorado Anschutz, Colorado State University, and the National Science Foundation (award 2201538). We thank Ben Aoki Sherwood and Nico Lang for providing valuable feedback during the writing stages of this work.

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

## A    ESTIMATORS OF INTRINSIC DIMENSION

### A.1    DISTANCE-BASED ESTIMATORS

**MLE (Levina & Bickel, 2005)** Let $z \in \mathbb{R}^D$ be an embedding (feature vector) and denote by radii $R_1(z) \leq \cdots \leq R_k(z)$ the Euclidean distances to its $k$ nearest neighbors. A broad family of intrinsic–dimension estimators rests on the assumption that, in a sufficiently small neighborhood of $z$, samples are drawn from an approximately uniform Poisson process on a $d$-dimensional manifold. Under this hypothesis one can write down a closed-form density for the joint distribution of the radii, which depends on $d$ only through the volume of a unit $d$-ball. The *Levina–Bickel MLE* (Levina & Bickel, 2005) estimator leverages this fact to define a local estimator

$$\widehat{d}_{\mathrm{MLE}}(z) \;=\; \left[ \frac{1}{k-1} \sum_{j=1}^{k-1} \ln \frac{R_k(z)}{R_j(z)} \right]^{-1},$$

whose global counterpart pools the local estimates using its harmonic mean across embeddings $z_i$,

$$\hat{d}_{\mathrm{global}} = \left( \frac{1}{n} \sum_{i=1}^{n} \frac{1}{\hat{d}_{\mathrm{MLE}}(z_i)} \right)^{-1} = \frac{n}{\sum_{i=1}^{n} \left[ \frac{1}{k-1} \sum_{j=1}^{k-1} \ln \frac{R_k(z_i)}{R_j(z_i)} \right]}.$$

**MOM (Amsaleg et al., 2018)** Under the same Poisson–ball model, conditioning on the outer radius $R_k(z)$ makes the inner neighbors approximately uniform inside the $d$-ball of radius $R_k(z)$. In higher

Table 2: **Intrinsic dimension compared to PCA and ICA as a function of embedding size.** (a) Global intrinsic dimension (ID) estimates for an image encoder (RCF) and a geospatial location encoder (SatCLIP, $L = 10$) are stable as the ambient embedding size $D$ increases. (b) For SatCLIP, PCA and ICA retain many more components as $D$ grows and closely track the ambient dimensionality, while ID and downstream task performance change stay relatively constant.

(a) ID stability across embedding sizes.

| Encoder | $D$ | MLE | FisherS | MOM | TLE |
|---------|-----|------|---------|------|------|
| RCF | 64 | 5.56 | 1.68 | 5.58 | 7.09 |
| | 128 | 5.57 | 1.68 | 5.48 | 7.11 |
| | 256 | 5.64 | 1.68 | 5.56 | 7.12 |
| | 512 | 6.32 | 1.64 | 5.23 | 7.10 |
| SatCLIP | 64 | 1.958 | 4.91 | 2.12 | 2.15 |
| | 128 | 1.958 | 5.00 | 2.10 | 2.13 |
| | 256 | 1.967 | 5.00 | 2.09 | 2.16 |
| | 512 | 1.956 | 5.00 | 2.10 | 2.16 |

(b) PCA/ICA components and downstream performance.

| $D$ | ID (MLE) | PCA 99% var. | ICA comps. | Air temp $R^2$ | Elev. $R^2$ | Pop. $R^2$ | Countries Top-1 | Biome Top-1 |
|-----|----------|--------------|------------|----------------|-------------|------------|-----------------|-------------|
| 64 | 1.958 | 36 | 64 | $95.9 \pm 0.18$ | $81.5 \pm 2.13$ | $78.4 \pm 0.51$ | $93.8 \pm 0.16$ | $92.0 \pm 0.47$ |
| 128 | 1.958 | 46 | 127 | $95.8 \pm 0.28$ | $81.8 \pm 0.45$ | $78.1 \pm 0.76$ | $94.1 \pm 0.17$ | $91.6 \pm 0.35$ |
| 256 | 1.967 | 84 | 256 | $95.1 \pm 0.15$ | $80.8 \pm 1.88$ | $79.0 \pm 1.03$ | $93.9 \pm 0.16$ | $92.23 \pm 0.25$ |
| 512 | 1.956 | 59 | 512 | $95.9 \pm 0.27$ | $81.2 \pm 1.52$ | $78.4 \pm 0.44$ | $94.4 \pm 0.00$ | $91.8 \pm 0.29$ |

dimensions more mass concentrates near the boundary, so the average inner radius moves closer to $R_k(z)$. Let

$$\bar{R}(z) \;=\; \frac{1}{k}\sum_{j=1}^{k} R_j(z).$$

The method of moments estimator equates the empirical ratio $\bar{R}(z)/R_k(z)$ to its model value and solves for $d$, which yields the local estimator

$$\widehat{d}_{\text{MOM}}(z) \;=\; \frac{\bar{R}(z)}{R_k(z) - \bar{R}(z)}.$$

Large $\widehat{d}_{\text{MOM}}(z)$ therefore corresponds to neighborhoods where most points lie close to the outer shell.

**TLE (Amsaleg et al., 2019).** Tight Local Estimation keeps the Levina–Bickel likelihood identity but extracts logarithmic spacing information from the full $k$-NN configuration, not only the ordered radii. Inside the ball of radius $R_k(z)$ it constructs pairwise geometric surrogates $S_{ij}(z)$ and $T_{ij}(z) \in (0, R_k(z)]$ that summarize how neighbors are positioned relative to one another. Degenerate or near–zero terms are discarded by a fixed threshold to stabilize the estimate. Schematically, the MLE log–ratios $\{\ln(R_k/R_j)\}$ are replaced by $\{\ln(R_k/S_{ij}), \ln(R_k/T_{ij})\}$ and combined through the same harmonic–mean aggregation to produce $\widehat{d}_{\text{TLE}}(z)$. The result is less sensitive to very small spacings while remaining driven by how tightly the neighborhood fills the $d$-ball.

**ESS (Johnsson et al., 2014).** Expected Simplex Skewness uses the shape of the neighbor constellation rather than only radial spacings. After mean–centering the $k$-NN patch at $z$, one forms vectors to neighbors and builds many small simplices from these vectors. A normalized "skewness" statistic $s(z)$ combines their volumes and edge lengths. For a uniform set of finite-neighbor samples around $z$ in a $d$-ball this statistic concentrates around a dimension–specific reference curve. The local estimate $\widehat{d}_{\text{ESS}}(z)$ is obtained by inverting the observed $s(z)$ against that curve, using linear interpolation between consecutive integer dimensions. Because ESS is driven mainly by relative geometry and angles, it tends to be less sensitive to mild density gradients while still reflecting curvature and noise through systematic changes in the neighbor shape.

## A.2 ANGLE-BASED ESTIMATORS

We provide a detailed explanation of the angle-based FisherS estimator below (Albergante et al., 2019).

Let $\{z_i\}_{i=1}^n \subset \mathbb{R}^D$ denote embeddings. The FisherS estimator starts from a concentration-of-measure assumption: in high dimensions, points sampled uniformly on a sphere tend to be almost orthogonal, so a typical point can be linearly separated from the rest by a Fisher discriminant[1]. FisherS quantifies how separable each point is from all others at a similarity margin $\alpha \in (0,1)$, then inverts a closed-form reference curve to recover an effective dimension.

Following Albergante et al. (2019), inputs are standardized to make the separability model comparable with a uniform $n$-sphere: (i) center $Z$ by subtracting the sample mean; (ii) reduce dimension using PCA, retaining components with eigenvalues at least $\lambda_{\max}/C$ (default $C=10$), which removes ill-conditioned directions; (iii) whiten the retained coordinates so the covariance is the identity; and (iv) project each vector to the unit sphere, $x_i := z_i/\|z_i\|$. These steps yield an array $X \in \mathbb{R}^{n \times d}$ with rows $x_i$.

For each $\alpha$ on a grid, we form the normalized Gram matrix $G = XX^\top$ and test, for every pair $(i,j)$ with $i \neq j$, whether

$$\langle x_i, x_j \rangle \; > \; \alpha \langle x_i, x_i \rangle .$$

Since $\|x_i\| = 1$, this reduces to $\langle x_i, x_j \rangle > \alpha$, that is, whether the cosine similarity exceeds $\alpha$. Point $x_i$ is called *inseparable* at level $\alpha$ if it has at least one neighbor whose similarity crosses this threshold. Empirically we record, for each $i$, the fraction

$$p_i(\alpha) \;=\; \frac{1}{n} \#\big\{\, j \neq i : \langle x_i, x_j \rangle > \alpha \,\big\}$$

and then average over points to obtain $\bar{p}(\alpha) = \frac{1}{n} \sum_i p_i(\alpha)$.

If points are sampled uniformly on the unit $n$-sphere, the mean inseparability probability at margin $\alpha$ is well-approximated by

$$\bar{p}_{\mathrm{sphere}}(\alpha; n) \;\approx\; \frac{(1 - \alpha^2)^{(n-1)/2}}{\alpha\sqrt{2\pi n}} .$$

Given the measured $\bar{p}(\alpha)$, FisherS solves for $n$ by inverting this expression with the Lambert $W$ function:

$$\hat{n}(\alpha) \;=\; \frac{\mathrm{W}\!\left(-\dfrac{\ln(1-\alpha^2)}{2\pi\,\bar{p}(\alpha)^2\,\alpha^2(1-\alpha^2)}\right)}{-\ln(1-\alpha^2)} .$$

Varying $\alpha$ yields a profile $\alpha \mapsto \hat{n}(\alpha)$; FisherS then picks a single global estimate by selecting the largest $\alpha$ for which $\bar{p}(\alpha) > 0$ (i.e., not all points are fully separable) and reporting $\hat{n}(\alpha^\star)$ for $\alpha^\star \approx 0.9 \max\{\alpha : \bar{p}(\alpha) > 0\}$. The method also supports pointwise ID by plugging $p_i(\alpha)$ into the same inversion at a chosen $\alpha$.

FisherS does not rely on nearest-neighbor radii. It asks instead: "At a given angular margin $\alpha$, how often do points fail to be linearly separable from the rest?" On a true $n$-sphere, this failure rate decays at a rate governed by $n$. If the dataset is more clustered or locally low-dimensional, inseparability persists to larger $\alpha$, inflating $\bar{p}(\alpha)$ and thus lowering the inferred $n$; if the dataset spreads more uniformly, $\bar{p}(\alpha)$ drops and the inferred $n$ rises. The PCA+whitening+sphere projection makes this comparison meaningful by matching the spherical reference.

## B HOW DO THE NUMBER OF NEAREST NEIGHBORS CHANGE GEOGRAPHIC INR ID?

In distance-based estimators, the neighbor count $k$ controls a bias–variance tradeoff. With small $k$, the estimate is anchored to an extremely local neighborhood of $z$, so it is sensitive to the true local geometry and density but suffers high variance because it averages (harmonic mean in (Levina &

---

[1]Fisher's linear discriminant is the single direction (a line) that best separates two groups by making their projected means far apart relative to their spread.

Table 3: **Global intrinsic dimensions of geographic INRs derived from SatCLIP (Klemmer et al., 2025), GeoCLIP (Cepeda et al., 2023), and CSP (Mai et al., 2023a)**. For all distance-based estimators, we use $k = 20$ nearest neighbors. Results shown for 6 sampling schemes, estimated by six total methods. Variation shows as $1\times$ standard deviation over 3 sub-sampling runs containing $N = 50{,}000$ points each.

| Scheme | Model | CorrInt | FisherS | MLE | MOM | TLE | TwoNN |
|---|---|---|---|---|---|---|---|
| Fibonacci | SatCLIP–L10 | 2.16 $\pm$0.02 | 5.94 $\pm$0.00 | 2.45 $\pm$0.05 | 2.15 $\pm$0.00 | 2.43 $\pm$0.02 | **44.14 $\pm$0.80** |
| | SatCLIP–L40 | 2.36 $\pm$0.02 | 9.95 $\pm$0.03 | 2.58 $\pm$0.05 | 4.68 $\pm$0.10 | 2.78 $\pm$0.03 | 12.80 $\pm$0.30 |
| | **GeoCLIP** | **22.70 $\pm$0.05** | **11.16 $\pm$0.10** | **13.17 $\pm$0.10** | **23.70 $\pm$0.30** | **13.20 $\pm$0.20** | **24.15 $\pm$0.08** |
| | CSP–FMoW | 2.50 $\pm$0.03 | 1.87 $\pm$0.01 | 6.28 $\pm$0.10 | 5.07 $\pm$0.08 | 7.02 $\pm$0.12 | 9.43 $\pm$0.20 |
| | CSP–iNat | 5.65 $\pm$0.06 | 0.92 $\pm$0.01 | 4.96 $\pm$0.08 | 6.34 $\pm$0.10 | 5.90 $\pm$0.10 | 5.07 $\pm$0.15 |
| Sphere | SatCLIP–L10 | 2.01 $\pm$0.02 | 5.94 $\pm$0.03 | 2.01 $\pm$0.02 | 2.12 $\pm$0.02 | 2.22 $\pm$0.02 | 2.00 $\pm$0.06 |
| | SatCLIP–L40 | 2.18 $\pm$0.02 | 9.89 $\pm$0.03 | 2.14 $\pm$0.03 | 4.51 $\pm$0.10 | 2.52 $\pm$0.02 | 2.02 $\pm$0.06 |
| | **GeoCLIP** | **20.79 $\pm$0.04** | **11.10 $\pm$0.01** | **11.50 $\pm$0.02** | **22.48 $\pm$0.03** | **12.08 $\pm$0.02** | **15.07 $\pm$0.08** |
| | CSP–FMoW | 2.50 $\pm$0.03 | 1.87 $\pm$0.01 | 5.43 $\pm$0.06 | 4.96 $\pm$0.06 | 6.33 $\pm$0.10 | 4.53 $\pm$0.15 |
| | CSP–iNat | 5.58 $\pm$0.05 | 0.92 $\pm$0.01 | 4.27 $\pm$0.05 | 6.15 $\pm$0.08 | 5.31 $\pm$0.08 | 3.19 $\pm$0.10 |
| Land | SatCLIP–L10 | 1.96 $\pm$0.00 | 5.00 $\pm$0.01 | 1.96 $\pm$0.00 | 2.02 $\pm$0.00 | 2.16 $\pm$0.00 | 1.98 $\pm$0.01 |
| | SatCLIP–L40 | 2.05 $\pm$0.00 | **8.08 $\pm$0.00** | 2.03 $\pm$0.00 | 2.39 $\pm$0.00 | 2.32 $\pm$0.00 | 1.99 $\pm$0.01 |
| | **GeoCLIP** | **12.07 $\pm$0.05** | 7.68 $\pm$0.10 | **11.22 $\pm$0.10** | **13.02 $\pm$0.20** | **11.53 $\pm$0.15** | **11.25 $\pm$0.50** |
| | CSP–FMoW | 2.16 $\pm$0.03 | 1.70 $\pm$0.01 | 5.18 $\pm$0.08 | 5.23 $\pm$0.08 | 6.25 $\pm$0.10 | 3.05 $\pm$0.12 |
| | CSP–iNat | 3.93 $\pm$0.05 | 0.92 $\pm$0.01 | 3.37 $\pm$0.06 | 4.65 $\pm$0.08 | 4.14 $\pm$0.08 | 2.70 $\pm$0.10 |
| Grid | SatCLIP–L10 | 1.64 $\pm$0.02 | 4.45 $\pm$0.00 | 2.00 $\pm$0.00 | 2.15 $\pm$0.00 | 2.22 $\pm$0.00 | 2.61 $\pm$0.01 |
| | SatCLIP–L40 | 1.72 $\pm$0.02 | **8.76 $\pm$0.02** | 2.15 $\pm$0.00 | 3.40 $\pm$0.00 | 2.56 $\pm$0.00 | 2.58 $\pm$0.03 |
| | **GeoCLIP** | **18.87 $\pm$0.08** | 7.08 $\pm$0.01 | **13.00 $\pm$0.00** | **22.22 $\pm$0.00** | **13.55 $\pm$0.01** | **15.42 $\pm$0.10** |
| | CSP–FMoW | 2.65 $\pm$0.03 | 1.81 $\pm$0.01 | 5.69 $\pm$0.08 | 5.25 $\pm$0.08 | 6.56 $\pm$0.10 | 5.86 $\pm$0.20 |
| | CSP–iNat | 5.57 $\pm$0.06 | 0.92 $\pm$0.01 | 4.64 $\pm$0.08 | 6.32 $\pm$0.10 | 5.63 $\pm$0.10 | 3.68 $\pm$0.10 |
| Naive | SatCLIP–L10 | 1.76 $\pm$0.02 | 4.47 $\pm$0.03 | 2.01 $\pm$0.02 | 2.12 $\pm$0.02 | 2.22 $\pm$0.02 | 2.01 $\pm$0.08 |
| | SatCLIP–L40 | 1.95 $\pm$0.02 | **8.93 $\pm$0.03** | 2.15 $\pm$0.03 | 4.53 $\pm$0.10 | 2.54 $\pm$0.02 | 2.02 $\pm$0.08 |
| | **GeoCLIP** | **18.02 $\pm$0.08** | 7.09 $\pm$0.00 | **12.08 $\pm$0.00** | **21.53 $\pm$0.03** | **12.76 $\pm$0.02** | **13.63 $\pm$0.07** |
| | CSP–FMoW | 2.68 $\pm$0.03 | 1.81 $\pm$0.01 | 5.60 $\pm$0.08 | 5.26 $\pm$0.08 | 6.53 $\pm$0.10 | 4.86 $\pm$0.15 |
| | CSP–iNat | 5.51 $\pm$0.05 | 0.92 $\pm$0.01 | 4.41 $\pm$0.08 | 6.24 $\pm$0.10 | 5.45 $\pm$0.10 | 3.20 $\pm$0.10 |
| Stratified | SatCLIP–L10 | 1.75 $\pm$0.01 | 4.47 $\pm$0.00 | 2.01 $\pm$0.00 | 2.26 $\pm$0.00 | 2.22 $\pm$0.00 | 2.00 $\pm$0.01 |
| | SatCLIP–L40 | 1.92 $\pm$0.01 | 8.91 $\pm$0.01 | 2.15 $\pm$0.00 | 2.49 $\pm$0.00 | 2.54 $\pm$0.00 | 2.01 $\pm$0.01 |
| | **GeoCLIP** | **17.93 $\pm$0.08** | **7.07 $\pm$0.01** | **12.06 $\pm$0.10** | **21.53 $\pm$0.03** | **12.75 $\pm$0.02** | **13.58 $\pm$0.08** |
| | CSP–FMoW | 2.67 $\pm$0.03 | 1.81 $\pm$0.01 | 5.59 $\pm$0.08 | 5.25 $\pm$0.08 | 6.53 $\pm$0.10 | 4.79 $\pm$0.15 |
| | CSP–iNat | 5.51 $\pm$0.05 | 0.92 $\pm$0.01 | 4.42 $\pm$0.08 | 6.24 $\pm$0.10 | 5.45 $\pm$0.10 | 3.22 $\pm$0.10 |

Bickel, 2005) and arithmetic mean in (Amsaleg et al., 2019; 2018)) over a few nearest neighbors. As $k$ increases, variance decreases by pooling more radii. The effective neighborhood increases and begins to mix in curvature, density gradients, and boundary effects. These departures from local homogeneity induce bias whose sign and magnitude depend on the estimator. For the Levina–Bickel MLE estimator specifically, which is derived under a local homogeneous Poisson model, the estimator is approximately unbiased when the model holds and $k$ is small to moderate, and its variance scales like $\approx d^2/(k-3)$ for fixed intrinsic dimension $d$ (Levina & Bickel, 2005); pushing $k$ larger suppresses variance further but also violates the locality assumptions, allowing global structure to pull the estimate upward or downward.

Across estimators, increasing $k$ changes the balance between local detail and global structure, and our results reflect this clearly. ESS grows rapidly with $k$, most strikingly for GeoCLIP: under Fibonacci sampling it rises from about 12 at $k{=}5$ to nearly 99 at $k{=}200$. This pattern is consistent with ESS aggregating more directions and longer chords as the neighborhood widens, which inflates the measured simplex skewness and maps to a higher effective dimension. MOM and TLE typically drop from $k{=}5$ to the $k{\approx}20$–50 range, then flatten or rebound slightly. The initial decrease suggests that adding a few more neighbors stabilizes local spacing statistics and trims extreme ratios, while very large neighborhoods begin to mix curvature and density variation, reintroducing upward bias. MLE is comparatively stable. For GeoCLIP it is often flat to gently increasing with $k$ and

Table 4: **Global geographic INR ID versus number of nearest neighbors ($k$) for distance-based estimators on `scikit-dimension`.** 100,000 points sampled with land, fibonacci, and spherical sampling.

(a) MLE (Levina & Bickel, 2005)

| Sampling | Encoder | k | | | | | |
|---|---|---|---|---|---|---|---|
| | | 5 | 10 | 20 | 50 | 100 | 200 |
| land | GeoCLIP | 12.781 | 12.690 | 11.234 | 10.327 | 11.935 | 15.090 |
| | CSP-FMoW | 4.050 | 4.770 | 5.169 | 5.109 | 4.736 | 4.245 |
| | CSP-iNat | 2.940 | 3.151 | 3.375 | 3.691 | 3.930 | 4.095 |
| | SatCLIP-L10 | 1.982 | 1.969 | 1.960 | 1.952 | 1.951 | 1.963 |
| | SatCLIP-L40 | 1.993 | 2.004 | 2.024 | 2.099 | 2.225 | 2.497 |
| fibonacci | GeoCLIP | 17.070 | 13.041 | 13.173 | 17.795 | 22.188 | 25.560 |
| | CSP-FMoW | 8.027 | 7.087 | 6.280 | 5.352 | 4.749 | 4.231 |
| | CSP-iNat | 4.787 | 4.792 | 4.964 | 5.269 | 5.457 | 5.588 |
| | SatCLIP-L10 | 4.490 | 2.610 | 2.445 | 2.207 | 2.161 | 2.170 |
| | SatCLIP-L40 | 4.747 | 2.765 | 2.576 | 2.847 | 3.852 | 5.497 |
| sphere | GeoCLIP | 12.301 | 11.047 | 11.494 | 15.979 | 20.590 | 24.393 |
| | CSP-FMoW | 5.158 | 5.483 | 5.432 | 5.016 | 4.591 | 4.151 |
| | CSP-iNat | 3.480 | 3.848 | 4.272 | 4.851 | 5.208 | 5.443 |
| | SatCLIP-L10 | 2.007 | 2.009 | 2.013 | 2.032 | 2.058 | 2.110 |
| | SatCLIP-L40 | 2.043 | 2.072 | 2.140 | 2.553 | 3.529 | 5.135 |

(b) MOM (Amsaleg et al., 2018)

| Sampling | Encoder | k | | | | | |
|---|---|---|---|---|---|---|---|
| | | 5 | 10 | 20 | 50 | 100 | 200 |
| land | GeoCLIP | 25.754 | 17.070 | 12.988 | 11.282 | 13.015 | 16.269 |
| | CSP-FMoW | 8.823 | 7.037 | 6.440 | 5.802 | 5.217 | 4.567 |
| | CSP-iNat | 5.752 | 4.493 | 4.250 | 4.396 | 4.643 | 4.886 |
| | SatCLIP-L10 | 3.389 | 2.511 | 2.211 | 2.058 | 2.017 | 2.017 |
| | SatCLIP-L40 | 3.407 | 2.565 | 2.296 | 2.248 | 2.392 | 2.804 |
| fibonacci | GeoCLIP | 26.250 | 15.685 | 14.756 | 19.302 | 23.700 | 27.114 |
| | CSP-FMoW | 13.145 | 8.991 | 7.191 | 5.808 | 5.069 | 4.476 |
| | CSP-iNat | 8.372 | 6.484 | 6.100 | 6.196 | 6.340 | 6.411 |
| | SatCLIP-L10 | 7.298 | 2.799 | 2.496 | 2.191 | 2.151 | 2.178 |
| | SatCLIP-L40 | 7.389 | 2.968 | 2.649 | 3.134 | 4.683 | 6.989 |
| sphere | GeoCLIP | 20.905 | 14.051 | 13.451 | 17.942 | 22.484 | 26.187 |
| | CSP-FMoW | 10.176 | 7.579 | 6.490 | 5.553 | 4.959 | 4.423 |
| | CSP-iNat | 6.980 | 5.651 | 5.541 | 5.877 | 6.147 | 6.306 |
| | SatCLIP-L10 | 3.411 | 2.551 | 2.262 | 2.133 | 2.120 | 2.161 |
| | SatCLIP-L40 | 3.508 | 2.658 | 2.453 | 3.006 | 4.522 | 6.802 |

(c) TLE (Amsaleg et al., 2019)

| Sampling | Encoder | k | | | | | |
|---|---|---|---|---|---|---|---|
| | | 5 | 10 | 20 | 50 | 100 | 200 |
| land | GeoCLIP | 21.665 | 14.764 | 11.547 | 10.218 | 11.669 | 14.278 |
| | CSP-FMoW | 7.851 | 6.630 | 6.236 | 5.720 | 5.197 | 4.602 |
| | CSP-iNat | 5.070 | 4.259 | 4.149 | 4.326 | 4.551 | 4.753 |
| | SatCLIP-L10 | 2.786 | 2.319 | 2.155 | 2.085 | 2.084 | 2.117 |
| | SatCLIP-L40 | 2.892 | 2.446 | 2.319 | 2.355 | 2.523 | 2.923 |
| fibonacci | GeoCLIP | 23.199 | 14.034 | 13.196 | 16.840 | 20.173 | 22.568 |
| | CSP-FMoW | 12.282 | 8.618 | 7.023 | 5.777 | 5.100 | 4.546 |
| | CSP-iNat | 7.756 | 6.209 | 5.898 | 5.962 | 6.049 | 6.057 |
| | SatCLIP-L10 | 3.937 | 2.498 | 2.427 | 2.273 | 2.279 | 2.340 |
| | SatCLIP-L40 | 5.858 | 2.970 | 2.784 | 3.305 | 4.779 | 6.884 |
| sphere | GeoCLIP | 18.149 | 12.514 | 12.064 | 15.750 | 19.257 | 21.906 |
| | CSP-FMoW | 9.241 | 7.206 | 6.330 | 5.521 | 4.985 | 4.488 |
| | CSP-iNat | 6.180 | 5.301 | 5.305 | 5.637 | 5.856 | 5.952 |
| | SatCLIP-L10 | 2.827 | 2.372 | 2.224 | 2.188 | 2.225 | 2.306 |
| | SatCLIP-L40 | 3.039 | 2.586 | 2.519 | 3.106 | 4.532 | 6.607 |

(d) ESS (Johnsson et al., 2014)

| Sampling | Encoder | k | | | | | |
|---|---|---|---|---|---|---|---|
| | | 5 | 10 | 20 | 50 | 100 | 200 |
| land | GeoCLIP | 12.043 | 25.661 | 29.229 | 30.517 | 37.955 | 50.230 |
| | CSP-FMoW | 5.621 | 7.107 | 7.523 | 7.239 | 6.570 | 5.718 |
| | CSP-iNat | 3.860 | 4.192 | 4.414 | 4.781 | 5.108 | 5.418 |
| | SatCLIP-L10 | 2.148 | 2.097 | 2.072 | 2.097 | 2.166 | 2.290 |
| | SatCLIP-L40 | 2.222 | 2.249 | 2.356 | 2.632 | 2.970 | 3.682 |
| fibonacci | GeoCLIP | 12.031 | 25.870 | 39.545 | 63.910 | 84.450 | 98.897 |
| | CSP-FMoW | 7.524 | 9.148 | 8.400 | 7.033 | 6.123 | 5.344 |
| | CSP-iNat | 5.158 | 5.878 | 6.179 | 6.590 | 6.848 | 7.005 |
| | SatCLIP-L10 | 2.726 | 2.326 | 2.293 | 2.326 | 2.438 | 2.622 |
| | SatCLIP-L40 | 2.852 | 2.753 | 2.868 | 4.075 | 6.557 | 10.122 |
| sphere | GeoCLIP | 11.801 | 24.231 | 34.033 | 54.676 | 74.804 | 91.335 |
| | CSP-FMoW | 6.269 | 7.630 | 7.492 | 6.698 | 5.987 | 5.292 |
| | CSP-iNat | 4.438 | 5.116 | 5.596 | 6.236 | 6.627 | 6.881 |
| | SatCLIP-L10 | 2.168 | 2.138 | 2.145 | 2.250 | 2.398 | 2.601 |
| | SatCLIP-L40 | 2.320 | 2.443 | 2.686 | 3.809 | 6.064 | 9.429 |

occasionally U-shaped on land sampling. For SatCLIP encoders all distance-based estimators are remarkably steady, hovering near 2–3 across $k$, which indicates a tightly regular local geometry in those embeddings. Sampling also matters: fibonacci and spherical layouts amplify sensitivity to $k$ for GeoCLIP and the CSP encoders, whereas land sampling tends to mute it. Overall, the table shows that methods that directly emphasize extreme interpoint ratios or long-range chords move the most with $k$, while estimators that average log-radii more evenly are less affected.

## C    EXPERIMENT DETAILS: INTRINSIC DIMENSION OF GEOSPATIAL IMAGE ENCODERS

For all image encoders displayed on Table 1, we compute intrinsic dimension (ID) on features extracted from the S2-100K Sentinel-2 multispectral image dataset (Klemmer et al., 2025). Models that accept multispectral inputs use all 13 Sentinel-2 bands; models that only support RGB receive bands B4/B3/B2 in that order. All inputs are resized/cropped to $224 \times 224$ pixels unless noted, and RGB models use standard ImageNet normalization. We estimate global ID from the resulting feature matrices using multiple estimators as in Table 1 (for distance-based methods, $k = 20$ nearest neighbors). Below, we list key experimental details for each image encoder.

**Random Convolutional Filters (RCF) (Rolf et al., 2021).**    We use training-free RCFs as a simple baseline: 13-channel inputs are convolved with a random filter bank, passed through a nonlinearity, and spatially pooled into a fixed-length descriptor. No external weights are loaded. The feature dimension is set to $512$ for accurate comparison with larger image encoders.

**CROMA (Fuller et al., 2023).** We evaluate the optical branch of CROMA's multimodal encoder (pre-trained on Sentinel-1/2 pairs). Inputs are the 12 optical Sentinel-2 bands (with an excluded B10 cirrus band), resized to $120 \times 120$ pixels, and normalized to $[0, 1]$ following official pre-trained model usage guidelines. Intrinsic dimension is calculated on the pooled optical embedding produced by the pre-trained encoder weights.

**DOFA (Xiong et al., 2024).** DOFA is a masked-autoencoder foundation model that conditions its patch embedding on channel wavelengths, enabling a single model to handle arbitrary band sets. We pass the 13 Sentinel-2 central wavelengths with the 13-band inputs and extract encoder features from the publicly released pre-trained weights.

**ScaleMAE (Reed et al., 2023).** ScaleMAE is a scale-aware masked autoencoder trained on RGB imagery with ground-sampling-distance cues and a multiscale reconstruction objective. We feed RGB tiles (B4/B3/B2), apply ImageNet normalization, and use the pre-trained encoder's feature output.

**ResNet baselines (He et al., 2016).** For multispectral ResNets, we use 13-band Sentinel-2 MoCo weights (ResNet-18/50) and extract the global-average-pooled penultimate features. Pre-trained Sentinel-2 weights are loaded with TorchGeo package (Stewart et al., 2025). For RGB variants, we use the corresponding Sentinel-2 RGB MoCo weights with B4/B3/B2 inputs. For ResNet-152, we rely on ImageNet-1K pre-trained weights (RGB only) and take the pooled backbone features.

**ViT-Small (multispectral) (Dosovitskiy et al., 2021).** We use the Sentinel-2 MoCo-pre-trained ViT-Small that accepts 13 bands. We run a forward pass to obtain patch tokens and mean-pool them (excluding any prefix/CLS token) to form a single feature vector per tile.

**AlphaEarth (Brown et al., 2025).** We extract embeddings from the AlphaEarth embedding API, which provides dense, pre-computed 64-dimensional feature vectors derived from Sentinel-2 satellite imagery at 10-meter spatial resolution. These embeddings are accessed through Google Earth Engine's image collection (GOOGLE/SATELLITE_EMBEDDING/V1/ANNUAL), with annual composites available from 2017 to 2024. Each embedding dimension represents learned features from AlphaEarth's foundation model trained on global Earth observation data. The results in Table 1 use the single-pixel embedding (no averaging over space) and downloads embeddings for 2024. The embeddings are queried using geographic coordinates (longitude, latitude).

Table 5: Global ID estimates of AlphaEarth embeddings versus embedding sampling buffer size (meters).

| Buffer (m) | FisherS | MLE | MOM | TLE |
|---|---|---|---|---|
| 20 | 4.67 | 7.80 | 9.48 | 8.61 |
| 100 | 4.43 | 7.23 | 8.78 | 8.04 |
| 200 | 4.40 | 6.96 | 8.45 | 7.77 |
| 500 | 4.37 | 6.54 | 7.97 | 7.36 |

To investigate how spatial context affects the intrinsic dimensionality of AlphaEarth embeddings, we employ a systematic buffer-based aggregation strategy. For each coordinate point, we extract AlphaEarth embeddings using circular buffers varying from 10 meters around the geographic coordinates to 500 meters (1,256 pixels approximately aggregated). Within each buffer zone, embeddings are aggregated using mean reduction. From Table 5, we interestingly observe that increasing buffer sizes mildly decrease the global ID of AlphaEarth embeddings across all four estimators.

**SINR (Cole et al., 2023a).** The SINR encoder from Cole et al. (2023a) was originally developed for global-scale species distribution modeling from presence-only observation data. The SINR model uses sinusoidal encodings of geographic coordinates as input and learns implicit neural representations to predict species presence across 47k species. We extracted 256-dimensional feature vectors from approximately 100,000 globally distributed locations in the S2-100K dataset from the pre-trained SINR model trained on 100 observations/species.

**TaxaBind (Sastry et al., 2025)** We evaluate both the satellite image and location encoders from the TaxaBind model. The satellite encoder is a CLIP ViT-B/16 model. To process the 13-band Sentinel-2 inputs, we first select the RGB bands (B4, B3, B2). These 3-channel images are then resized to $224 \times 224$ and normalized using standard ImageNet statistics. The TaxaBind location encoder uses an identical model architecture as GeoCLIP (Cepeda et al., 2023) consisting of an Equal Earth projection, an RFF transform fed to an MLP.

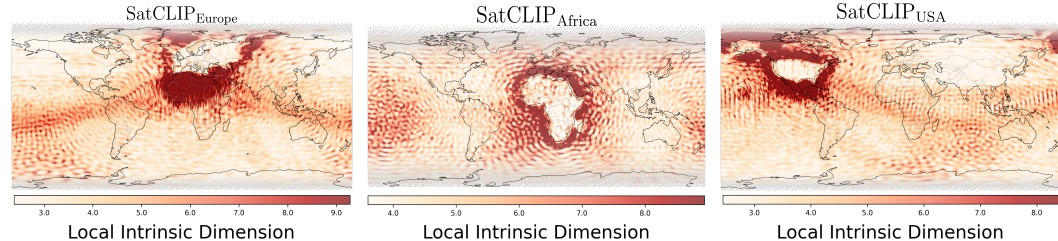

Figure 7: **Local intrinsic dimension can identify pre-training dataset coverage of local models.** We visualize local ID maps with the MLE estimator using $k = 100$ nearest neighbors. 100,000 geographic coordinates are sampled with the Fibonacci sampling scheme.

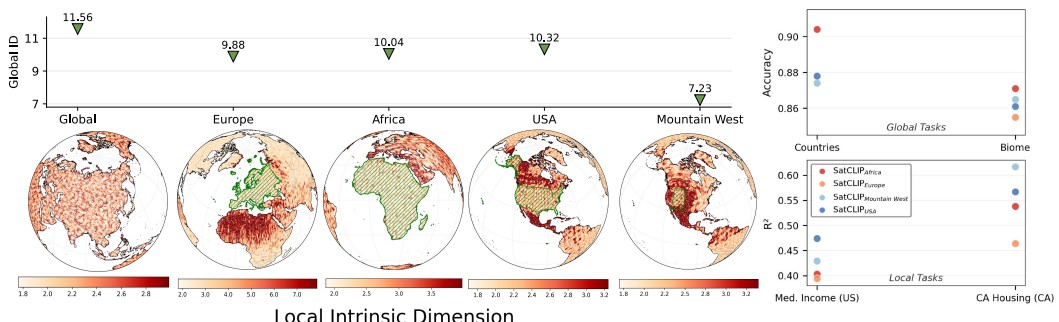

Figure 8: **Effect of pre-training dataset coverage on local and global geographic INR ID**: We generate several variants of the S2-100K datasets constrained to (i) continent-scale, (ii) country-scale, and (iii) region-scale boundaries, and train SatCLIP with $L = 40$ Legendre polynomials on each regional dataset. (Top) Global geographic INR ID for each regional location encoder. (Bottom) Local ID with the MLE estimator using $k = 100$ neighbors. Marked green outlines indicate spatial coverage of pre-training dataset. (Right) Performance of regional SatCLIP location encoders on (top) global and (bottom) local evaluation tasks.

**DINO-SAT (Siméoni et al., 2025).** We evaluate the DINOv3 (ViT-7B/16) model, which was pre-trained on the SAT-493M dataset of high-resolution Maxar imagery. To apply this model to our 13-band Sentinel-2 dataset, we first select the visual-spectrum bands (B4, B3, and B2) to create a 3-channel RGB image. This RGB image is then passed to the model's official Hugging Face AutoImageProcessor, which handles the resizing (to 224x224) and normalization that the model expects. We use the output CLS token as the final feature vector. Although the model was pre-trained on a different sensor (Maxar) at a different spatial resolution, this application is a valid test of the model's generality. We are assessing the model's capability as a powerful, general-purpose visual feature extractor on a new, medium-resolution satellite domain while retaining a fair comparison to other image encoders in Table 1.

## D    EXPERIMENT DETAILS: EFFECT OF SPATIAL RESOLUTION AND SPATIAL COVERAGE ON GEOGRAPHIC INR ID

We detail our experimental framework that measures changes in the geographic INR ID with changing spatial resolution of the location encoder. We also include an additional experiment where we show the relationship between local and global geographic INR ID and spatial coverage of the pre-training data.

### D.1    MODIFYING THE SPATIAL RESOLUTION OF GEOGRAPHIC LOCATION ENCODERS

We study the effect of increasing spatial resolution in SatCLIP on ID by pre-training variants that differ only in the spherical–harmonic band–limit $L$ (higher $L$ allows finer spatial variation), while holding the location–embedding dimension and all other hyperparameters fixed. After pre-training,

we compute location embeddings for $N=100{,}000$ *land–only* coordinates (uniform on $S^2$ then restricted to land) and report the global intrinsic dimension using the FisherS estimator.

GeoCLIP aligns images with GPS using a CLIP-style contrastive objective and a continuous location encoder built from hierarchical random Fourier features (RFF) (Cepeda et al., 2023). As the public codebase provides inference and pre-trained models rather than end-to-end pre-training,[2] we fine-tune *only* the location encoder on an image-to-GPS retrieval loss, freezing the CLIP image tower and the temperature (logit scale), with early stopping on validation loss (patience= 20 epochs). To modulate spatial resolution we (i) densify the RFF hierarchy by increasing the number of log-spaced frequency levels $M$ while holding $\sigma_{\min}, \sigma_{\max}$ fixed, and (ii) append one or two higher-frequency RFF branches to extend $\sigma_{\max}$ while keeping existing branches frozen. Note that in the hierarchy-level-sweep with $M$, we instantiate the location encoder from scratch for each $M$ (no warm-start), avoiding inherited bias from a previous hierarchy layout. When adding higher RFF frequency branches, however, we copy weights for overlapping branches and freeze them, so any ID change can be attributed to the new high-frequency branches. After fine-tuning, we estimate global intrinsic dimension with the FisherS estimator on embeddings of uniformly sampled land-only coordinates, and report all results under this land-only sampling.

We study the effect of increased spatial resolution on ID in Sphere2Vec (Mai et al., 2023b) and Space2Vec (Mai et al., 2020) by modifying the multi-scale resolution parameter $S$. All encoders are instantiated with trained weights from the MOSAIKS nightlights task (Rolf et al., 2021). For **Sphere2Vec**, we modulate spatial resolution by sweeping the number of scales $S$ (TorchSpatial: `freq`)—$S \in \{2, 4, 8, 16, 32\}$. We keep the output width fixed (`spa_embed_dim=256`). We apply this to SphereM, SphereM+, SphereC, and SphereC+. Increasing $S$ (and decreasing `min_radius`) adds higher-frequency components, yielding finer spatial detail. For **Space2Vec** we control resolution via either the *grid* sampling density (e.g., $W \times H \in \{(180, 90), (360, 180), (720, 360), (1440, 720)\}$) or the number of theory frequencies $S \in \{2, 4, 8, 16, 32\}$. We sample $N=100{,}000$ coordinates uniformly on $S^2$, compute embeddings (deduplicating exact repeats), and estimate *global* intrinsic dimension with the FisherS estimator. We report ID as a function of the resolution knob for each encoder family in Figure 5.

---

[2]`https://github.com/VicenteVivan/geo-clip`

### D.2 ADDITIONAL EXPERIMENT: MODIFYING THE SPATIAL COVERAGE OF PRE-TRAINING DATA

Geospatial models with strong global results can still underperform in specific regions; recent studies show that locally trained models or locally fine-tuned variants can match or exceed the accuracy of globally trained systems in the same area, underscoring a local–global divide in practice (Rolf et al., 2024).

Motivated by this, we ask how the *spatial coverage* of pre-training data shapes the geometry of location encoders as seen through their global and local IDs. We construct region–constrained pre-training corpora (continent, country, sub-region) from Sentinel-2 using the Google Earth Engine API and train SatCLIP location encoders with fixed capacity and optimization budget, varying only the geographic extent. We then estimate a global FisherS ID on embeddings computed on a global scale and a *local* point-wise ID (MLE) that we visualize on orthographic maps.

Table 6: Regional Sentinel-2 datasets used for local Sat-CLIP pre-training (tiles at 10 m, 256×256).

| Region | # Samples |
|---|---|
| United States | 100,000 |
| Europe | 100,000 |
| Asia | 100,000 |
| Africa | 100,000 |
| France | 50,000 |
| Mountain West | 50,000 |
| Denver Metro | 20,000 |

To study how pre-training *coverage* shapes geographic INRs, we build region–specific Sentinel-2 datasets and pre-train regional SatCLIP models, then measure the ID of location embeddings derived from each local encoder. Regions are defined by Natural Earth polygons; for each region we sample point locations (random or stratified; cached for determinism) and retrieve COPERNICUS/S2_SR_HARMONIZED imagery via Google Earth Engine, selecting the least-cloudy scene in a fixed window between 2023-01-01–2024-12-31. Regions we create local datasets for, along with the number of Sentinel-2 tiles queried are displayed in Table 6.

Around each point we extract a multi-band 256×256 tile at 10 m, stack bands $\{B1, \ldots, B12\}$ into a single GeoTIFF, and resample if needed to preserve exact size. Per region we release the images, an `index.csv` of (lon, lat), and the scripts used to generate them. For regional pre-training, we keep the SatCLIP pre-training hyperparameters fixed and set the spherical-harmonic band-limit to $L{=}40$ (location-embedding dimension and all other hyperparameters held constant). The only factor that varies across runs is the pre-training dataset. For ID evaluation, we feed the trained location encoder (no classifier) with $N$ coordinates sampled globally and report global FisherS ID.

From Figures 7 and 8, as pre-training coverage narrows from global to continental to regional, the global ID decreases. The local ID exhibits a complementary and highly diagnostic pattern: values remain modest *inside* the training extent but rise sharply in its immediate geodesic neighborhood, forming anisotropic "halos" that effectively delineate the pre-training footprint. This finding adds further evidence that the intrinsic dimension can identify the spatial coverage of pre-training data.

## E VALIDATING OUR ESTIMATES OF INTRINSIC DIMENSION

We verify the reliability of intrinsic-dimension (ID) estimators for geographic INRs by constructing a testbed where the "ground truth" ID is effectively known. Pope et al. (2021) validated ID estimators in vision by creating synthetic image manifolds with a known upper bound on ID via GAN latent dimension, then checking whether estimators recover the expected scaling. This established a baseline of estimator reliability for image data. For spherical INRs without additional information augmentation (such as those done in Klemmer et al. (2025); Cepeda et al. (2023); Mai et al. (2023a)), the underlying representation geometry can be described by the 2-sphere ($S^2$). We validate whether common ID estimators behave sensibly under sphere-respecting positional encodings and simple learned mappings (where the "true" ID should remain $\approx 2$).

Our brief experimental framework first measures the intrinsic dimension of raw geographic coordinates. This is a toy example where the ambient (extrinsic) dimension is equal to the intrinsic dimension. We increase the difficulty by passing raw longitude–latitude coordinates through a spherical–harmonics (SH) positional encoder (Rußwurm et al., 2024). Concretely, the encoder maps $(\lambda, \varphi) \in S^2$ to the concatenated vector of real SH basis values $\Phi_L(\lambda, \varphi) = \left[ Y_{\ell,m}(\lambda, \varphi) \right]_{\ell \leq L, \, -\ell \leq m \leq \ell} \in \mathbb{R}^{\sum_{\ell=0}^{L}(2\ell+1)}$, where $Y_{\ell,m}$ are the (real) spherical harmonics (Eqs. (3)–(5) in (Rußwurm et al., 2024)). While arbitrary nonlinear transformations can alter ID, maps that

are smooth and locally invertible (bi-Lipschitz, or more generally $C^1$ with full-rank Jacobian—i.e., immersions) do not. They reparameterize the underlying manifold without adding degrees of freedom. The spherical–harmonic encoder $\Phi_L$ and its compositions with linear layers or smooth pointwise nonlinearities (e.g., Siren) satisfy these conditions generically (Jacobian rank = 2 almost everywhere on $S^2$), so the intrinsic dimension should remain 2.

Table 7 shows that the FisherS estimator is consistently closest to the ground–truth ID = 2 across all stages (raw, SH, SH+linear, SH+Siren), yielding the smallest MAE overall. In contrast, the distance–ratio estimators MLE/MOM, and TLE are biased high—especially after applying SH and learned mappings—indicating greater sensitivity to the embedding and neighborhood construction.

In Fig. 10, interestingly, both the MLE and FisherS ID estimators produce a latitude-based striping with the raw geographic coordinates that persists when these geographic coordinates are passed to the SH positional encoder. A plausible explanation is periodicity and coordinate singularities of longitude–latitude: although the points lie on $S^2$, $(\lambda, \phi)$ imposes a rectangular chart with a discontinuity at the international date line ($\lambda = \pm 180°$) and singular behavior at the poles. As a result, locations that are geodesically close on the sphere can appear far apart (or vice versa) when distances and neighborhoods are constructed in raw $(\lambda, \phi)$ space across the wrap and near the poles. Non-linearities introduced by a linear or siren-network change these distinct latitude-based striping patterns for both the MLE and FisherS estimator. We find additional evidence here that local ID maps point to properties of the positional encoding functions used to construct the geographic INRs: a spherical harmonic positional encoding followed by a linear network creates crowding and striping at the poles (column SH + LINEAR NN in Figure 10) which is removed completely when we use a SirenNet (column SH + SIRENNET).

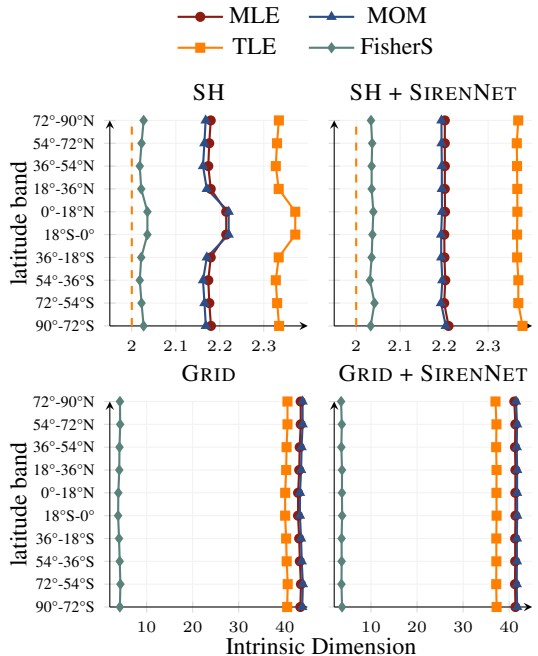

Figure 9: **Latitudinal variation of ID estimators with a Spherical Harmonics (SH) and Grid encoding of geographic co-ordinates**. The FisherS estimator is the most robust to latitudinal effects and different encodings of geographic coordinates.

Lastly, we find strong evidence from Figure 9 that the FisherS is most robust to latitudinal variation of intrinsic dimension, and is significantly more robust when comparing ID results between geographic INRs constructed with different positional encoders. The Grid positional encoding (employed in Mai et al. (2023a)) causes distance-based estimators of ID to present high values of ID, but FisherS stays close to the true ID of 2.

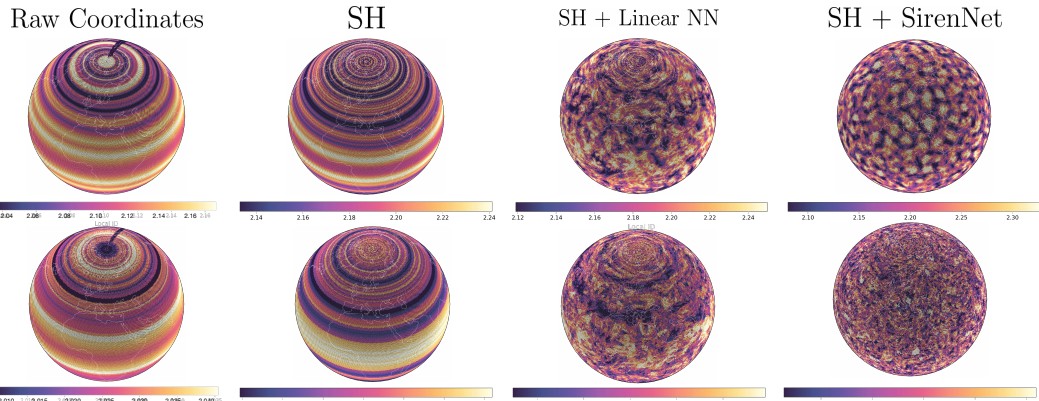

Figure 10: **Verifying local ID estimates of embeddings derived from** $S^2$**.** Top row shows local ID estimates of the MLE estimator (distance-based) and bottom row shows estimates from the FisherS estimator (angle-based). Columns indicate the local ID plots with (i) raw geographic coordinates $(\lambda, \phi)$ (ii) pure spherical-harmonic-based embeddings (SH) of $(\lambda, \phi)$, and spherical harmonic embeddings passed through a randomly initialized (iii) linear network and (iv) Sinusoidal representation network (Sitzmann et al., 2020; Rußwurm et al., 2024).

Table 7: Intrinsic dimension estimates for spherical data (true ID = 2.0) with (i) raw geographic coordinates $(\lambda, \phi)$ (ii) pure spherical-harmonic-based embeddings of $(\lambda, \phi)$, and spherical harmonic embeddings augmented with a randomly initialized (iii) linear network and (iv) Sinusoidal representation network (Sitzmann et al., 2020; Rußwurm et al., 2024). Values show mean ± standard deviation. Best values (closest to 2.0) for each stage are shown in bold.

| Stage | MLE | MOM | TLE | FisherS |
|---|---|---|---|---|
| Raw Coordinates | $2.113 \pm 0.056$ | $2.097 \pm 0.069$ | $2.038 \pm 0.039$ | $\mathbf{2.024 \pm 0.008}$ |
| Pure SH | $2.189 \pm 0.032$ | $2.183 \pm 0.046$ | $2.344 \pm 0.034$ | $\mathbf{2.025 \pm 0.011}$ |
| SH + Linear | $2.186 \pm 0.040$ | $2.179 \pm 0.055$ | $2.340 \pm 0.039$ | $\mathbf{2.026 \pm 0.008}$ |
| SH + SIREN | $2.201 \pm 0.075$ | $2.194 \pm 0.090$ | $2.367 \pm 0.056$ | $\mathbf{2.036 \pm 0.073}$ |
| **MAE** | 0.172 | 0.163 | 0.272 | **0.028** |

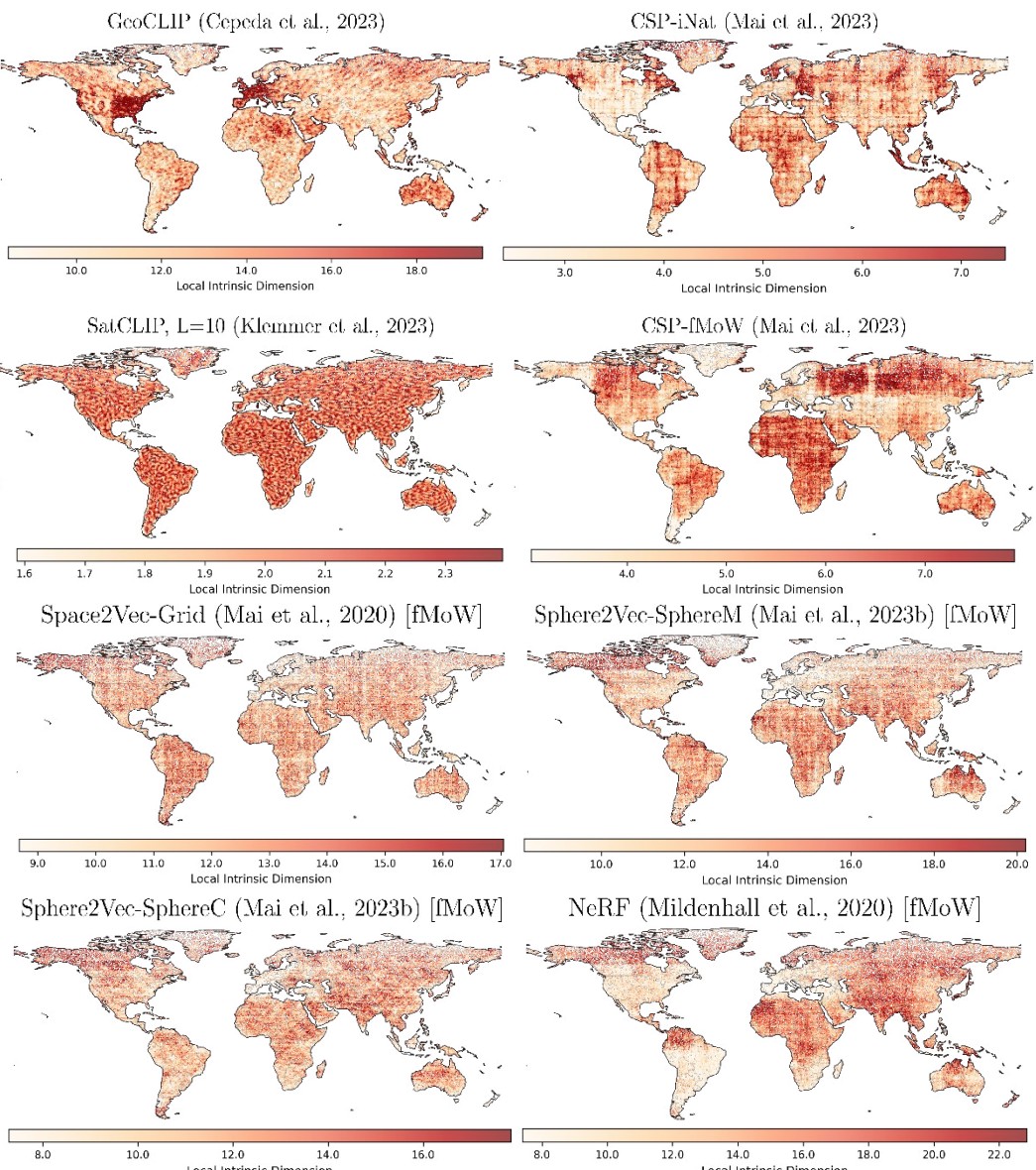

Figure 11: **Full Plot: Local intrinsic dimension of geographic INRs reveal spatial artifacts.** MLE estimator with $k = 100$ nearest neighbors used. All Sphere2Vec, Space2Vec, and NeRF models are trained on the functional map of the world dataset (Christie et al., 2018) with supervised learning.

