# OpenReview forum: "Measuring the Intrinsic Dimension of Earth Representations"
_ICLR.cc/2026/Conference — ICLR 2026 Poster_

### Official Review · Reviewer_Ayws · 2025-10-26

**Soundness:** 2
**Presentation:** 2
**Contribution:** 2
**Rating:** 2
**Confidence:** 4

**Summary:**

The paper presents techniques to measure intrinsic dimension and degree of task-alignment for various geospatial location and image encoders. The paper compares various global and local measures of intrinsic dimension. The paper conducts various experiments comparing the ID of location and image encoder against downstream task performance. This could prove valuable to the community for analyzing the informativeness of geospatial embeddings.

**Strengths:**

1. The paper is well motivated and makes a novel contribution by analyzing the intrinsic dimension of various geospatial location and image encoders and comparing it with downstream task performance.
2. Geospatial embeddings are increasingly being used to address various geospatial tasks. The paper presents a methodology to evaluate the "goodness" of these embeddings generated using existing geospatial encoders. This methodology can be very useful for the community in future.

**Weaknesses:**

1. It is difficult to comprehend clearly the methods used for computing global and local IDs. Theory is included in the supplementary material which is very useful but I highly suggest moving some important mathematical theory into the main paper.
2. Although measuring IDs of geospatial models an interesting direction, the paper falls short of several experiments that could help strengthen the paper:
* How do IDs relate to PCA/ICA and geospatial embedding visualization? It might be interesting to discuss how IDs can be used for possibly better visualization of geospatial embeddings than maybe using PCA/ICA which are not spatially explicit.
* One important experiment is missing that compares ID with the embedding dimension for each model architectures. The paper should discuss how the intrinsic dimension is related to the overall dimensionality of the embeddings (for the same model architecture and training regime). For example, if GeoCLIP is retrained with an embedding dimension of 768 or 1024, how would the intrinsic dimension change? Would it increase, decrease or remain the same. The same goes for image encoders. How would the intrinsic dimension vary when ScaleMAE is trained with different embedding dimensions.
* For measuring the IDs of image encoders, the authors should consider comparing the IDs across different spatial resolution of satellite images. **Comparing the IDs of image encoders using low resolution Sentinel-2 imagery is not enough to conclude the global ID of GeoCLIP approaches to that of various image encoders**. The authors should also experiment with high resolution imagery as that will possibly increase the IDs of image encoders.
* Concrete measures for task-alignment need to be shown. The authors could compute a mutual information estimate such as CLUB [1] between the geospatial embeddings and downstream task variables and compare it against the ID values to truly understand if IDs are correlated with downstream task performance.
* Can ID measure be made more task-specific since tasks vary in spatial frequency. Currently, ID seems to be independent of task and only dependent on the embedding model.
* **How do IDs relate to other downstream tasks that go beyond memorization** especially where the location encoders are used as a geo-prior such as satellite image generation or fine-grained image classification? **I believe the results demonstrated in the paper can easily be gamed by overfitting on the task, the authors need to show some analysis on tasks that are more challenging such as generation.**
* It should be interesting to see IDs for non-geospatial image encoders such as Jepa or Dinov2. Recently, webscale version of Dinov3 has shown to outperform the satellite-only Dinov3 model on various geospatial downstream tasks.
* Limited location and image encoders are compared. There are recent task-aligned location encoder models such as SINR, TaxaBind and Climplicit. There are multimodal image encoders such as AnySat and Galileo and Range. It will be interesting to see the ID analysis on such models.

**Suggestion**

I appreciate the authors for exploring the IDs of various geospatial models. The idea and motivation behind the paper is novel and will surely be useful for the community. However, the paper requires a major revision as the **paper makes strong claims without a comprehensive and sound experimental setup**. I would suggest the authors to include the above suggested experiments to improve the paper. The paper should clearly state concrete applications of computing the IDs of geospatial models. The paper also has typos which need to be fixed. The paper could be structured better with more theory in the main paper and better explanations for technical terms. I am definitely willing to increase my score once the authors have addressed my concerns.

**Questions:**

Please see weaknesses.

---

> ### Author Response · Authors · 2025-11-19
> **Response by the Authors (1/3)**
>
> Thank you very much for your detailed evaluation of our work, along with the valuable suggestions for improvement\!
>
> ## Responses to Weaknesses
>
> ### \[W1\] I highly suggest moving some important mathematical theory into the main paper.
>
> We fully agree with this suggestion, but were limited to 9 pages for the paper submission and decided to focus on our results rather than background on the previously proposed ID estimators. With the post-rebuttal 10-page limit, we have now added additional theory in Section 2.2.1, with new text colored in blue.
>
> ### \[W2b\] The paper should discuss how the intrinsic dimension is related to the overall dimensionality of the embeddings (for the same model architecture and training regime).
>
> We find that across all the ID estimators used in Table 1, ID does not change significantly with an increase in ambient embedding size $D$. The table below summarizes this result, where we vary $D$ for an image embedding model (RCF) and a location encoder model (SatCLIP) while fixing all other training and architectural components.
>
> | Encoder | Emb. Size (D) | MLE | FisherS | MOM | TLE |
> | :---- | :---- | :---- | :---- | :---- | :---- |
> | **RCF** | 64 | 5.56 | 1.68 | 5.58 | 7.09 |
> |  | 128 | 5.57 | 1.68 | 5.48 | 7.11 |
> |  | 256 | 5.64 | 1.68 | 5.56 | 7.12 |
> |  | 512 | 6.32 | 1.64 | 5.23 | 7.10 |
> | **SatCLIP** | 64 | 1.958 | 4.91 | 2.12 | 2.15 |
> | (L=10) | 128 | 1.958 | 5.00 | 2.10 | 2.13 |
> |  | 256 | 1.967 | 5.00 | 2.09 | 2.16 |
> |  | 512 | 1.956 | 5.00 | 2.10 | 2.16 |
>
> ####
>
> ### \[W2a\] How do IDs relate to PCA/ICA and geospatial embedding visualization?
>
> We find that both PCA and ICA are less suitable than intrinsic dimension for geospatial embeddings. Following our experimental setup for our previous response, when we increase $D$, the number of components retained rises significantly for PCA and ICA (for ICA, the number of retained components is almost always equal to the ambient embedding size\!).
>
>
>
> | Emb. Size (D) | ID (MLE) | PCA (99% Var.) | ICA (Sig. Comps.) | Air Temp ($R^2$) | Elevation ($R^2$) | Population ($R^2$) | Countries (Top-1) | Biome (Top-1) |
> | :---- | :---- | :---- | :---- | :---- | :---- | :---- | :---- | :---- |
> | 64 | **1.958** | 36 | 64 | 95.9 $\\pm$ 0.18 | 81.5 $\\pm$ 2.13 | 78.4 $\\pm$ 0.51 | 93.8 $\\pm$ 0.16 | 92.0 $\\pm$ 0.47 |
> | 128 | **1.958** | 46 | 127 | 95.8 $\\pm$ 0.28 | 81.8 $\\pm$ 0.45 | 78.1 $\\pm$ 0.76 | 94.1 $\\pm$ 0.17 | 91.6 $\\pm$ 0.35 |
> | 256 | **1.967** | 84 | 256 | 95.1 $\\pm$ 0.15 | 80.8 $\\pm$ 1.88 | 79.0 $\\pm$ 1.03 | 93.9 $\\pm$ 0.16 | 92.23 $\\pm$ 0.25 |
> | 512 | **1.956** | 59 | 512 | 95.9 $\\pm$ 0.27 | 81.2 $\\pm$ 1.52 | 78.4 $\\pm$ 0.44 | 94.4 $\\pm$ 0.00 | 91.8 $\\pm$ 0.29 |
>
>
> When evaluated on five different classification and regression downstream tasks across five random seeds, we also find that increasing $D$ does not meaningfully increase downstream task performance. While PCA/ICA are good preliminary visualization tools in spatial representation learning, our finding is that they fail to serve as a reliable metric of information content and task-alignment for Earth Representations. This reinforces our statement on PCA/ICA in Section 2.2. We thank you for prompting us to study this, and have added this result in Appendix Table 2.
>
> ### \[W2c\] The authors should also experiment with high resolution imagery instead of only using low-resolution Sentinel-2 imagery.
>
> Sentinel-2 imagery is the highest resolution (10m Ground Sampling Distance (GSD)) imagery available which is (i) free-to-use and publicly available, (ii) achieves global coverage, and (iii) recent and actively updated. Thus, we focus on embeddings from Sentinel-2 multispectral imagery due to the importance of being able to sample evenly across the globe for our analysis of global ID. All image encoders (except DINOv3-Sat) listed in Table 1 contain Sentinel-2 in their pre-training modalities, ensuring a fair comparison between image and location encoders.
>
> We would like to emphasize that our goal in this work is to characterize the intrinsic dimension of Earth representations *as they are currently published / released*. Embeddings from high-resolution 3-channel RGB imagery can possibly have a higher intrinsic dimension than lower-resolution 12-channel imagery, but we find that these questions are out of scope for our research objective: We do not aim to establish an upper-bound on the intrinsic dimension of a given model architecture and training setup under higher resolutions of pre-training data. However, we believe that future work that quantifies this would provide valuable insight.

---

> > ### Author Response · Authors · 2025-11-19
> > **Response by the Authors (2/3)**
> >
> > ### [W2d] Concrete measures for task-alignment need to be shown (ex: Mutual Information estimate such as CLUB between the geospatial embeddings and downstream task variables).
> >
> > We agree that mutual information (MI) is a natural information-theoretic notion of task alignment. Our paper, however, adopts a deliberately geometric notion of task alignment: in Section 2, we define task alignment as the extent to which a simple downstream head can compress an implicit neural representation onto a low-dimensional, target-aligned manifold, and in Section 4, we quantify this via the ID of task-head’s activations across multiple downstream tasks. While utilizing CLUB would be an interesting direction, the paper’s authors themselves emphasize that MI estimation in high-dimensional continuous spaces (such as Earth representations) is challenging.
> >
> > We note that our definition of task-alignment mirrors those in prior work on deep networks, where low ID in the final hidden layer reliably predicts generalization performance. We have clarified this geometric notion of “task alignment” in Section 3.1 to avoid confusion with mutual-information–based definitions.
> >
> > ### [W2e] Can ID measure be made more task-specific since tasks vary in spatial frequency?
> > Our goal in this work is to develop an unsupervised, task-agnostic, model-agnostic measure of information content in geographic INRs. The fact that global ID is independent of any particular downstream label is therefore a valuable property: it allows us to compare and diagnose Earth representations before committing to specific tasks or training heads. At the same time, we do study task-specific structure by measuring the ID of task-head activations for multiple downstream problems, which captures how much of the INR’s capacity is actually used for a given target. We agree that designing explicitly task- or frequency-specific ID metrics (e.g., by re-weighting local ID estimates at different spatial scales for a particular task) would be an interesting complementary direction, but it is orthogonal to our central contribution of establishing ID as a general, label-free diagnostic for Earth representations.
> >
> > ### [W2f] How do IDs relate to other downstream tasks that go beyond memorization?
> > In this work we focus on introducing ID–based metrics of representativeness and task alignment, and on analyzing how these relate to performance on standard downstream uses of geographic INRs, rather than on optimizing the hardest possible tasks. We therefore use supervised regression and classification benchmarks (climate, socioeconomic, land-cover tasks) that are widely adopted to evaluate location encoders and come with simple, well-defined generalization metrics (RMSE, $R^2$, accuracy), instead of generative settings where evaluation is substantially more complex and often relies on subjective or model-specific criteria. We agree that extending ID-based analysis to settings such as location-conditioned satellite image generation or fine-grained classification with generative priors is an interesting and complementary direction and we have now mentioned this explicitly as future work in Section 5. However, this lies outside the scope of our central contribution of establishing ID as a simple, task-agnostic diagnostic for Earth representations.

---

> > > ### Author Response · Authors · 2025-11-19
> > > **Response by Authors (3/3)**
> > >
> > > ### [W2g] It should be interesting to see IDs for non-geospatial image encoders such as Jepa or Dinov2. Recently, webscale version of Dinov3 has shown to outperform the satellite-only Dinov3 model on various geospatial downstream tasks.
> > >
> > > Our experimental framework is intentionally defined for any model that produces embeddings in $\mathbb{R}^{D}$, and it can in principle be applied directly to large-scale SSL encoders such as JEPA, DINOv2, or web-scale DINOv3 (line 88-91 in our paper). In the revised manuscript, we have added DINOv3 with a 7B ViT trained on the SAT-493M dataset. As mentioned in the last paragraph of our introduction, we focus on geographic INRs because their underlying domain geometry is explicit: inputs lie on the sphere $S^2$, which lets us cleanly separate the known 2D manifold from the learned information content above it and analyze spatial patterns in local ID, rather than treating the input space as an abstract Euclidean domain. We plan on performing a larger-scale benchmark study of ID estimates across SSL models, including non-geospatial encoders in future collaborative work.
> > >
> > > ### [W2h] Limited location and image encoders are compared. There are recent task-aligned location encoder models such as SINR, TaxaBind and Climplicit. There are multimodal image encoders such as AnySat and Galileo and Range. It will be interesting to see the ID analysis on such models.
> > >
> > > We have modified Table 1 with ID estimates from the task-aligned SINR and TaxaBind location encoders, the recently introduced multi-modal AlphaEarth foundation model, TaxaBind’s satellite image encoder, and the DINOv3-ViT7B pretrained on the Sat-493M satellite image dataset. Furthermore, Figure 3 is augmented with both the SINR location encoder and the TaxaBind location encoder. These changes are on our revised manuscript in a blue font color, and add to 18 total image and location encoders benchmarked in our study.
> > >
> > > A general takeaway is that task-aligned Earth representations such as SINR (ID $\approx 3.19$) and TaxaBind (ID $\approx$ 3.39) have lower global ID values compared to models that generate general-purpose embeddings such as SatCLIP, AlphaEarth, DINOv3-Sat. This aligns with and strengthens our findings in Figure 3a and 3b.
> > >
> > > ### The paper should clearly state concrete applications of computing the IDs of geospatial models.
> > >
> > > Thank you for this suggestion. We have modified the discussion and conclusion to expand on the applications of computing ID for Earth representations. These applications include (i) model selection via label-free evaluation at the pre-training stage, (ii) an unsupervised proxy for downstream model performance, (iii) a regularization and early-stopping criterion for large-scale training of SSL models, and (iv) a principled way to resolve geographic biases that can inform data quantification and model error attribution.

---

### Official Review · Reviewer_1jrM · 2025-10-30

**Soundness:** 3
**Presentation:** 4
**Contribution:** 3
**Rating:** 8
**Confidence:** 3

**Summary:**

The paper presents a study on the intrinsic dimensionality of geographic implicit neural representations (INRs), quantifying how much real information these learned location embeddings actually capture. The authors present some interesting findings including that global ID estimates of geographic INRs are competitive with those of large-scale image encoders, and that that global ID correlates positively with downstream performance when measured on frozen, pre-trained models (representativeness), but correlates negatively when measured in the activation space of supervised models (task-alignment).

Overall, I find this work quite interesting and valuable, as their methodology offers a principled way to measure information content in location embedding techniques beyond traditional downstream performance metrics.

**Strengths:**

[S1] The paper is the first to examine the intrinsic dimension of location embeddings, providing valuable insights into the representational limits and strengths of popular approaches. The study is especially useful given the large and growing number of works on location embeddings in GeoAI.

[S2] The paper is well written and easy to follow. The methodology for estimating intrinsic dimension is clearly explained, and the results give interesting insights, including how intrinsic dimension varies with model architecture, input modalities, and spatial resolution.

[S3] The work is clearly written, well-organized, and thoroughly evaluated across a diverse range of models and experimental cases.

**Weaknesses:**

[W1] The TwoNN estimator is used to measure task alignment, with low ID values observed for task-specific representations. Can the authors elaborate on whether this could be an artifact of estimator bias due to non-uniform data, rather than actual compression?

[W2] We observe notable differences in Global ID estimates across different estimators, especially for SatCLIP (e.g., SatCLIP-L40: 8.08 for FisherS vs. 2–2.5 for MLE, TLE, and TwoNN). The authors primarily focus on FisherS, which consistently gives much higher values, but the reasons for this discrepancy are not fully discussed. Is such a large gap expected in practice? Can the authors provide further empirical or theoretical justification for prioritizing FisherS over other estimators for Global ID?

**Questions:**

[1] Please see W1 & W2.
[2] I am wondering if the study can generally extended to study encoders of geospatial objects such as [a] [b] [c]

[a] "Towards general-purpose representation learning of polygonal geometries." GeoInformatica 2023.

[b] "Poly2Vec: Polymorphic Fourier-Based Encoding of Geospatial Objects for GeoAI Applications." ICML 2025.

[c] "Geo2Vec: Shape-and Distance-Aware Neural Representation of Geospatial Entities." arXiv 2025.

---

> ### Author Response · Authors · 2025-11-19
> **Response by the Authors**
>
> Thank you very much for your careful analysis of our work and the insightful comments.
>
> ## Response to Weaknesses
>
> ### \[W1\] Can the authors elaborate on whether \[the low ID values from TwoNN on task-specific representations\] could be an artifact of estimator bias due to non-uniform data, rather than actual compression?
>
> Thank you for raising this great point. Indeed, it has been shown that estimators of ID can provide biased values due to non-uniform sampling of the underlying manifold [1]. This is why we chose downstream tasks that have global spatial coverage of data (see Figures 3 and 4) to not introduce this bias in the ID estimators. To further account for variability caused by non-uniform sampling of geographic coordinates, we calculate ID values over 6 different sampling schemes (detailed in Appendix Table 3), with results indicating small differences between the global sampling schemes. While it is impossible to completely rule out the effect of estimator bias in our ID calculations, we believe that testing on globally distributed downstream tasks and introducing metrics of variability across diverse sampling schemes limits the degree to which bias would influence our final ID estimates.
>
> ### \[W2\] Can the authors provide further empirical or theoretical justification for prioritizing FisherS over other estimators for Global ID?
>
> As mentioned in Section 2.2, the angle-based FisherS estimator is robust to spatial variabilities that are ubiquitous in Earth representations with changing terrains, coastlines, and administrative boundaries. Unlike the FisherS estimator, distance-based estimators (MLE, TLE, MOM) read these spatial variabilities  as changes in dimensionality, which makes them less suitable for global-scale comparisons. However, as mentioned in Section 3, it is this sensitivity to spatial variability that make distance-based estimators suitable for a local estimate of ID (Figure 2). In Appendix E, we empirically validated the suitability of FisherS for global ID using a controlled, synthetic experiment where we calculate the intrinsic dimension of raw geographic coordinates and simple non-linear transforms of these coordinates. Our design choice of the TwoNN estimator to measure “task-alignment”, i.e., the ID of the task-head in activation space, is motivated by literature in [2], where the authors use a similar experimental setup and ID estimator to measure the correlation between ID and generalization performance. We have revised the manuscript to detail these motivations and mathematical differences between estimators in the main text.
>
> **[1]** Levina, Elizaveta, and Peter Bickel. "Maximum likelihood estimation of intrinsic dimension." *Advances in Neural Information Processing Systems* 17 (2004).
>
> **[2]** Ansuini, Alessio, et al. "Intrinsic dimension of data representations in deep neural networks." *Advances in Neural Information Processing Systems 32* (2019).
>
> ## Responses to Questions
>
> ### [Q1] Can this study be extended to study encoders of geospatial objects such as Poly2Vec, Geo2Vec, etc?
>
> Thank you for the great question. Yes, our study of ID is generally applicable to any encoder that produces embeddings in $\\mathbb{R}^D$. This includes encoders of most geospatial objects including vector, polygonal, and polyline geometries that produce a high-dimensional embedding vector.

---

### Official Review · Reviewer_zZ97 · 2025-10-31

**Soundness:** 3
**Presentation:** 3
**Contribution:** 3
**Rating:** 8
**Confidence:** 4

**Summary:**

This paper presents a novel research on the intrinsic dimension (ID) of geographic Implicit Neural Representations (INRs). While the measures used to quantify ID already exist in literature, it is important to bring the concept to the community of spatial representation learning. The authors also conducted comprehensive analysis on how ID affects the downstream performance of geographic INRs, revealing that ID can be a good index of representation quality.

**Strengths:**

1. It is important to introduce the concept of ID to the community of spatial representation learning. Unlike image or text embedding where there is **compression** of information, spatial representation such as location encoding usually embeds very low dimensional data (e.g. two-dimensional latitudes and longitudes) into high dimensional spaces, which is not a compression but an expansion. The quality of the expansion, i.e., whether the embedding manifold maintains useful topological information, is a critical concern -- by the curse of dimensionality, higher dimensions tend to make embeddings less distinguishable and less informative.

2. Local ID and global ID allow fine-grained analysis of the embedding manifold.

3. The correlation analysis between ID and downstream task performance is very useful. The positive/negative correlations can be used to guide spatial representation learning or to select appropriate embedding methods for specific downstream tasks.

**Weaknesses:**

1. The linear correlations fitted in Figure 3 and Figure 4 can be misleading. From the authors' perspective, they only need to show that the ID and model performance are positively/negatively correlated; the correlation does not need to be linear. For example, the subplots in Figure 3a look more quadratic/logarithmic than linear.

2. Table 1 can not be used to fairly compare the ID of baseline models. The models evaluated are trained on very different datasets and downstream tasks. For example, GeoCLIP having very high ID, from my experience, owes a lot to the spatial bias of the MP16 dataset it was trained on. This is also shown in Figure 2. In order to make the comparison between numbers meaningful, it is better to train different models on the same dataset(s).

**Questions:**

1. The non-linear correlations in Figure 3 and Figure 4 may have a theoretical implication. In the 4th chapter of https://escholarship.org/uc/item/5bs589v5, it is mathematically proven that the average information content in the samples drawn within a given geographical region decreases logarithmically when there is spatial dependency.

2. Can you train some baseline models on a spatially balanced dataset (e.g. OSV5M) and compare their ID? This is more useful to prove that ID can be used to quantify the quality of an embedding method.

3. Do you have any idea why in Figure 5, the Space2Vec method has such large increase in ID compared to Sphere2Vec variants?

---

> ### Author Response · Authors · 2025-11-19
> **Response by the Authors**
>
> Thank you for the positive evaluation of our work and the constructive comments.
>
> ## Responses to Weaknesses
>
> ### [W1] The linear correlations fitted in Figure 3 and Figure 4 can be misleading. The correlation does not need to be linear.
>
> Thank you for this observation. Indeed, the correlations in Figure 3a do not seem to be linear (but Figs 3b and 4 could be linear). Our main message in this plot was to demonstrate (generally) a positive or negative correlation. To keep it simple, we chose a normal scale and a linear line as a visual aid for the reader to follow the direction of slope.
>
> In the now-revised manuscript, we have decreased the visibility and grayed these lines to better emphasize the data itself rather than the linear fit.
>
> ### [W2] In order to make the comparison between numbers in Table 1 meaningful, it is better to train different models on the same dataset(s).
>
> We would like to highlight multiple results in our work that calculate the ID of Earth representations in a controlled setting. In Table 1, we calculate the ID of the CSP location encoder with a fixed model architecture, but two distinct pre-training datasets (fMoW and iNaturalist2018). In Figure 5, We retain the same pre-training dataset per-model, and only modify the spatial resolution of SatCLIP, GeoCLIP, Sphere2Vec, and Space2Vec models. In Figure 6, we retain an identical model architecture while modifying the pre-training input modalities in a controlled manner. Additionally, in Appendix Figure 7, we fix SatCLIP’s model architecture while modifying the spatial coverage of its pre-training dataset. These results are examples where we ablate one component and observe meaningful changes in the intrinsic dimension that align with our core findings: that the ID is a principled measure of representativeness and task-alignment of Earth representations.
>
> In this context, we would like to highlight that our goal in this work is to characterize the intrinsic dimension of a broad range of current Earth embeddings as they are used by practitioners and to provide ID-based analysis and interpretability tools for future embedding approaches.
>
> ## Responses to Questions
>
> ### [Q1] The non-linear correlations in Figure 3 and Figure 4 may have a theoretical implication.
>
> Thank you very much for pointing this reference to us and informing us about the logarithmic decrease in information content amidst spatial autocorrelation. This interpretation indeed sounds valid from our results in Figure 3 and Figure 4. However, we are unable to access the article posted as it is “under embargo until April 30, 2026”. We believe it would be valuable to cite this article in our paper once we can access it and validate the relevance to our work.
>
> ### [Q2] Can you train some baseline models on a spatially balanced dataset (e.g. OSV5M) and compare their ID?
>
> Thank you for the suggestion to incorporate OSV5M into our experimental framework. We however notice that the OVS5M dataset is not spatially balanced over landmass, as indicated in Figure 3 of the OVS5M paper, and misses samples in large contiguous parts of Northern Africa, South America, and Asia, caused by the availability of street-view imagery in sparsely-populated, arid areas. In Figure 2, we see the direct effects of spatial coverage reflected in the local intrinsic dimension estimates: SatCLIP, which is pre-trained on the spatially-balanced S2-100K dataset (as indicated in Figure 3 of the SatCLIP paper) observes significantly fewer spatial artifacts compared to GeoCLIP, which, as mentioned, is pre-trained on a spatially-imbalanced MP-16 dataset with higher coverage in the United States and Europe. This finding is reinforced in Appendix Figure 7 where we intentionally ablate the S2-100K dataset to be spatially imbalanced over specific regions on Earth.
>
> ### [Q3] Do you have any idea why in Figure 5, the Space2Vec method has such a large increase in ID compared to Sphere2Vec variants?
>
> This is an excellent question, and one that has been raised during earlier development of this work. Our thoughts on why this might be the case: Space2Vec’s positional encoding is built from many separate grid patterns at different frequencies. When we add higher-frequency scales, the network effectively receives a lot of new, almost independent “directions” to use, and the measured ID increases significantly. Sphere2Vec uses a DFS PE that is built to preserve distances on $S^2$. Higher-frequency components mostly refine the same underlying spherical surface instead of opening up new directions. This keeps the embeddings closer to a manifold closer to the ID of $S^2$.

---

### Official Review · Reviewer_F7dp · 2025-11-01

**Soundness:** 3
**Presentation:** 4
**Contribution:** 4
**Rating:** 4
**Confidence:** 5

**Summary:**

This paper presents a framework for quantifying the intrinsic dimension of Earth representations and studying the correlation between ID with the model performance on downstream tasks.

**Strengths:**

1. The  author proposed to use Intrinsic Dimension (ID) as a new metric to quantify the representativeness and task-lignment of the Earth representation. It gives us a way to interpret the location encoders and how the design can impact the model performance
2. A thorough analysis has been carried out to analysis the relation between ID and the task performance in the context of Earth presentation learning.

**Weaknesses:**

Although I enjoy reading this paper, there are some weaknesses I need to point out:
1. In the main results table (Table 1), the authors compared different pretrained location encoders with the global ID metric. However, different location encoders have different designs, different pretraining objectives, and different pretraining datasets and modalities. It is very hard to see any pattern here. A controlled experiment is needed in which the type of location encoders, the pretraining objectives, and the pretrained datasets need to be ablated to see how different parts impact the ID and performance of the location representations.
2. Figure 6 has a similar problem: 3 models use different data modalities, different architectures, and different pretraining strategies. A controlled experiment is needed here.

**Questions:**

1. "Global IDs of geographic INRs are similar to that of embeddings derived from image encoders". However, compared with image encoders, location encoders have much less learnable parameters. Does this result indicate that the ID depends more on the dataset and task characteristics (e.g., spatial distribution of the class labels) instead of the model?
2. Figrue 4 shows that "Lower global ID in activation space of supervised models corresponds to higher task performance". Why the results from location embeddings and activation features show a reverse pattern? "past work, that found lower ID indicates more concentrated, linearly separable structure and thus better generalization" Can you explain this in details?

---

> ### Author Response · Authors · 2025-11-19
> **Response by the Authors**
>
> Thank you for your careful analysis of our work.
>
> ## Responses to Weaknesses
>
> ### [W1] Different location encoders have different designs, different pretraining objectives, and different pretraining datasets and modalities. It is very hard to see any pattern here.
>
> We consider the ability of intrinsic dimension to capture true information content present in Earth representations regardless of model architecture design, pre-training objectives, and pre-training dataset to be a major strength of ID as an unsupervised evaluation metric. This provides a post-hoc ID-based analysis tool that is model and training data agnostic and can be used even if model and training data are unknown, such as in the case of the new AlphaEarth foundation model.
>
> We would like to point out that we conduct experiments that systematically vary the spatial resolution (Figure 5) and input modalities (Figure 6) used to train location encoders, and we see explicit patterns; the effects are directly visible in the encoded information content of the location embeddings quantified through both its ID and downstream task performance.
>
>
> ### [W2] Figure 6 has a similar problem: 3 models use different data modalities, different architectures, and different pretraining strategies. A controlled experiment is needed here.
>
> Figure 6 in our paper does indeed use the same model architecture, pre-training strategy, spatial distribution of pre-training data, and hyperparameters with the only change being in the number of input modalities supplied to the model. All these models are the SatCLIP image-location contrastive learning model trained on the multi-modal MMEarth dataset. The purpose of this experiment is to study the effect different input modalities have on ID estimates, given other factors (architecture, loss) are retained in a controlled setting. We are happy to clarify further.
>
> ## Responses to Questions
>
> ### [Q1]  Does the result [in Table 1] indicate that the ID depends more on the dataset and task characteristics (e.g., spatial distribution of the class labels) instead of the model?
>
> Thank you for the insightful question. We would like to clarify that **our estimates of ID in Table 1 are independent of the spatial distribution of class labels**. All the geographic co-ordinates and imagery sampled to generate ID estimates in this Table originate from the spatially-balanced, globally available S2-100K dataset (sampled approximately uniform over landmass) proposed in SatCLIP. This allows for comparisons between models to be fair. In this controlled setting, we find that both image and location encoders encode similar amounts of information in its embeddings, as measured through their IDs.
>
> ### [Q2] Why do the results from location embeddings and activation features show a reverse pattern?
>
> This is an excellent observation, and highlights the core contribution of our work. Our results build on the seminal work of Ansuini et al. (2019) and Pope et al. (2021). Specifically, Ansuini et al. (2019) find that lower ID of the last hidden layer of standard vision models trained with supervised learning on common image classification datasets is inversely correlated with test performance. Similarly, Pope et al. (2021) find that image datasets with lower IDs show better generalization and sample-complexity results. Our work mirrors this finding through results displayed in Figure 3b and Figure 4 of our paper. In these results, we confirm that lower ID embeddings and activations after task-specific finetuning, which we term task-alignment, correlates with better downstream performance. Note that in Figure 3b, we follow the same experimental setup, and use the same ID estimator (TwoNN) as used in Ansuini et al. (2019).
>
> In Figure 3a, our results show that high ID of raw location representations produced by self-supervised learning is positively correlated with task performance. This is because unlike the supervised learning setup in Ansuini et al. (2019), self-supervised learning training paradigms generate embeddings with the explicit goal of global, general-purpose use, and representativeness. In these cases, a high ID is good for a variety of downstream tasks, as it separates data in embedding space across many dimensions. The supervised loss during fine-tuning then collapses away all irrelevant patterns to only the few that are relevant to one particular downstream task. Thus, our work adds valuable context to the long-studied ID-generalization relationship by showing that embeddings learned with self-supervised learning exhibit a positive ID-generalization correlation.
>
> **[Ansuini et al. (2019)]** Ansuini, Alessio, et al. "Intrinsic dimension of data representations in deep neural networks." *Advances in Neural Information Processing Systems* 32 (2019).
>
> **[Pope et al. (2021)]** Pope, Phil, et al. "The Intrinsic Dimension of Images and Its Impact on Learning". *International Conference on Learning Representations* (2021).

---

### Author Response · Authors · 2025-11-20
**General Response by the Authors to all Reviewers**

We thank all reviewers for their insightful questions and feedback on our work.

We are particularly encouraged by the consensus that our work is “novel” (Reviewers zZ97, Ayws, 1jrM), “well-motivated” (Reviewer Ayws), “useful” (Reviewers Ayws, zZ97, 1jrM), and the positive feedback that our work is “well-written, easy to follow, and well-organized” (Reviewer 1jRM) with our experimental setup on measuring the ID of geographic implicit neural representations described as “thorough” (Reviewer F7dp) and “comprehensive” (Reviewers zZ97, 1jrM). While the concept of intrinsic dimension has existed in classical information theory since the early 60s, and in deep learning for close to two decades, we are grateful for the reviewers for noting that this the first attempt to introduce this principled evaluation metric to the spatial representation learning community and location embeddings (Reviewers F7dp, zZ97, 1jrM) especially amidst their growing popularity and increased use (Reviewers Ayws, 1jrM).

We have responded to individual reviewers’ comments, and are confident that we can address the remaining questions and critiques during the remainder of the discussion period. Motivated by the feedback, we have made changes to our now revised manuscript with changes marked in blue font. We are available to assist if any questions remain.

---

### Comment · Area_Chair_QYoG · 2025-11-28
**Discussions**

Dear Reviewers,

The authors have given their rebuttals to your reviews. Could you please express your opinions?
Many thanks,
AC

---

### Author Response · Authors · 2025-12-03
**General Response by Authors at the end of the discussion period**

We would like to thank the reviewers and Area chairs for participating in the discussion phase. During the discussion phase, we highlight a score increase by Reviewer Ayws from 2 to 4 on November 21st after our posted response. We are available for any questions and/or clarifications from the AC through the end of the discussion period, and thank them for their effort in considering our work for publication at ICLR.

---

### Meta-Review · Area_Chair_GM9U · 2026-01-07

**Summary:**

The submission analyzes the intrinsic dimension of INR representations of geographic coordinates, giving insights into how different models represent geographic data.  It is an interesting application of dimensionality analysis to probe the empirical behavior of geographic models.

**Reviewer Concerns:**

Reviewers were concerned about interpretable patterns from diverse models.  The authors responded that the goal is to provide methodologies for understanding models in a very generic way.  Reviewer Ayws has many detailed questions, which the authors systematically address in their response.

**Reviewer Scores:**

The reviewers were split, with two recommending clear acceptance, and two recommending rejection.  The rebuttal addresses specific concerns, and the application is quite interesting.  The methodology is easily transferable to other models, and fits well with ICLR themes.  The response to Reviewer Ayws is quite extensive and systematically addresses questions in detail.

---

### Decision · Program_Chairs · 2026-01-26

Accept (Poster)